# Matching amino acids membrane preference profile to improve activity of antimicrobial peptides

Shanghyeon Kim [1], Jaehoo Lee[1], Sol Lee[1], Hyein Kim[1], Ji-Yeong Sim[1], Boryeong Pak[1], Kyeongmin Kim[2] &
Jae Il Kim [1,3✉]

Antimicrobial peptides (AMPs) are cationic antibiotics that can kill multidrug-resistant bacteria via membrane insertion. However, their weak activity limits their clinical use. Ironically, the cationic charge of AMPs is essential for membrane binding, but it obstructs membrane insertion. In this study, we postulate that this problem can be overcome by locating cationic amino acids at the energetically preferred membrane surface. All amino acids have an energetically preferred or less preferred membrane position profile, and this profile is strongly related to membrane insertion. However, most AMPs do not follow this profile. One exception is protegrin-1, a powerful but neglected AMP. In the present study, we found that a potent AMP, WCopW5, strongly resembles protegrin-1 and that the match between its sequence and the preferred position profile closely correlates with its antimicrobial activity. One of its derivatives, WCopW43, has antimicrobial activity comparable to that of the most effective AMPs in clinical use.

[1] School of Life Sciences, Gwangju Institute of Science and Technology, Gwangju 61005, Republic of Korea. [2] Department of Microbiology, School of Medicine, Kyungpook National University, 680 Gukchaebosangro, Jung-gu, Daegu 41944, Korea. [3] Pilot Plant, AnyGen, Gwangju, Technopark, 333 Cheomdangwagiro, Buk-gu, Gwangju 61008, Republic of Korea. ✉email: jikim@gist.ac.kr

Epidemic outbreaks of multidrug resistant (MDR) bacteria threaten human health. Especially, Gram-negative multidrug-resistant (MDR) bacteria (*K. pneumonia*, *A. baumannii*, *P. aeruginosa*) are the most urgent threat to human health[1,2]. These organisms readily acquire pan-drug resistance because their outer membranes block entry and/or facilitate the export of most antibiotics[1]. To overcome this situation, antimicrobial peptides (AMPs) are being considered as a potentially effective treatment for MDR bacterial infections[3,4]. Most AMPs are cationic and amphipathic α-helical peptides with affinity for anionic bacterial membranes, into which they insert and generate pores that disrupt the membrane[3,5,6]. This property underlies the efficacy of AMPs against MDR bacteria. For example, the outer membrane-targeting AMP colistin is used as a last resort antibiotic against gram-negative MDR bacteria, despite its known nephrotoxicity[2]. However, after decades of use, colistin-resistant Gram-negative bacteria have appeared and are spreading worldwide[7]. So far, no viable substitute for colistin has been reported. At present, even the most potent AMPs are not sufficiently active within the necessary minimal inhibitory concentration (MIC) range (≤4 µg/ml) defined by Clinical and Laboratory Standards Institute (CLSI) guidelines[4,8].

The cationic charge on AMPs is necessary for membrane binding, but it inhibits membrane insertion because charged molecules cannot efficiently insert into the hydrophobic center of a cell membrane[9–11]. Although the AMP–pore ensemble model[5] and cation–π interaction[12] suggest that AMP–AMP oligomerization would enable partial charge neutralization mediated by an interaction between cationic charged amino acids and aromatic ring amino acids, polar and helical AMP oligomers would struggle against the energy penalty imposed by insertion into the narrow hydrophobic gap between lipid tails[13] (Supplementary Fig. 1). The insertion of helical transmembrane proteins is assisted by enzymes and ATP[14], but that is not the case with AMP oligomers.

Energetically, the centers of membranes prefer peptides with carbon chain amino acids (Leu, Val) for insertion, whereas the surfaces of membranes prefer peptides with cationic amino acids (Arg, Lys), and the interfaces between membrane surfaces and centers prefer amino acids with aromatic rings[9–11] (Trp, Phe).

This preference profile for certain amino acids at certain membrane positions (referred to as hydrophobic profile complementarity[9] or partition coefficient[10]) is closely related to the membrane insertion mechanism of AMPs, as these amino acids are the most critical components of AMPs[5,6,15]. However, designing AMPs with optimized preferred position profiles is difficult because most AMPs are secondary amphipathic α-helices whose structures cannot be matched to preferred positions. Only primary (mostly β-strand) amphipathic AMPs such as protegrins-1 can match preferred membrane positions[16–19] (Fig. 1a, b).

Protegrin-1 is a potent AMP with a breakpoint MIC[16,20,21], and two protegrin-1-inspired AMPs (murepavadin[22] and iseganan[16,23]) have entered clinical trials. However, before protegrin-1-derived AMPs can be considered as next-generation antibiotic candidates, several critical problems must be overcome. These include their dependence on albumin and acetic acid for in vitro activity[20,21] (Supplementary Table 3); their unique β-strand structures;[15,24,25] the unpredictable and controversial effects of their disulfide bonds;[16,26–28] their toxicities;[15,16,24,25] their ability to form amyloid-like fibrils[15,29] and their low solubility (Supplementary Table 4).

AcWL5 (Acetyl-WLLLLL), the shortest transmembrane β-strand peptide consists of two essential AMP components, Trp and Leu[30–32]. In addition, the amino acid sequence of the dimerized AcWL5 peptide perfectly matches the preferred membrane positioning (Fig. 1)[30–32]. However, AcWL5 lacks the cations for membrane binding need by a potential AMP.

We previously generated an insect AMP (coprisin) derivative, termed CopW[33,34]. In this study, we developed a CopW derivative, WCopW5, which has a sequence similar to those of both protegrin-1 and AcWL5 and exhibits a CLSI breakpoint MIC (Table 1 and Supplementary Table 1). The membrane-interaction parameters and MICs of the tested WCopW5 derivatives (total, 72 derivatives) strongly suggest that amino acid sequence matching for the preferred membrane position is closely related to antimicrobial activity. The most effective AMP, WCopW43, exhibited in vitro antimicrobial activity comparable to that of the most effective AMPs in clinical use (colistin and daptomycin) and was effective in vivo in mouse models.

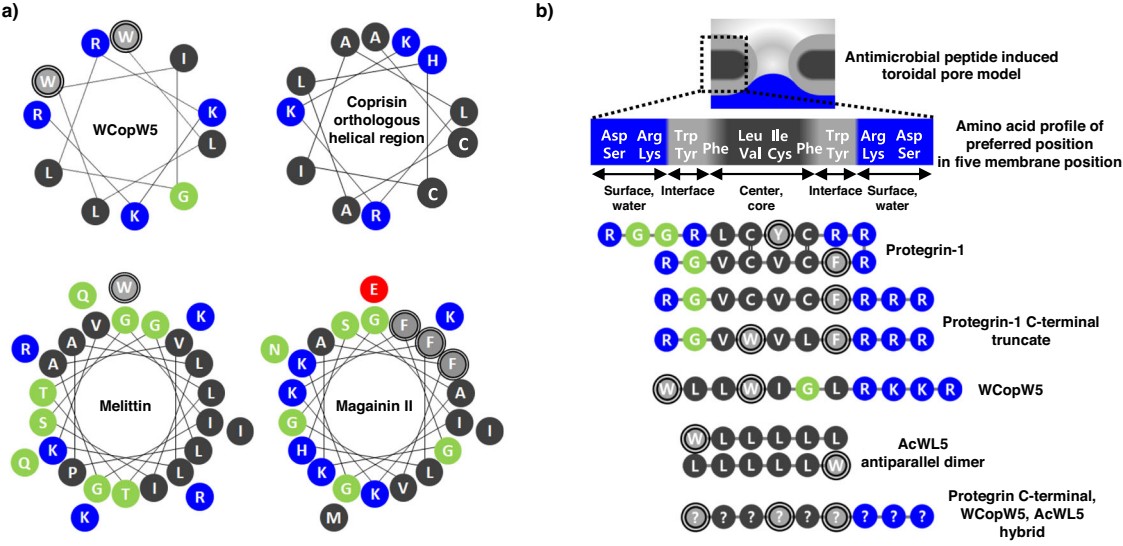

**Fig. 1 Predicted amphipathy of antimicrobial peptides. a** Secondary amphipathic α-helical model and **b** primary amphipathic β-strand model of AMPs. For simplicity, the membrane is shown as a simple electrostatic five-slab model. For easier viewing, the exact thickness of the membrane and chirality of the D-form amino acids are omitted and shown in supplementary figures[30,35] (Supplementary Figs. 1 and 2). WCopW5 is more similar to the primary amphipathic β-strand model of AMPs.

**Table 1 Sequence and activity of WCopW-derived AMPs.**

| Componud | Sequence[a] | | | | | | | | | | | | Net charge | MW | Minimum inhibitory concentrations (μM)[b] | | | |
|---|---|---|---|---|---|---|---|---|---|---|---|---|---|---|---|---|---|---|
| | | | | | | | | | | | | | | | MDR _P. aeruginosa_ | MDR _A. baumannii_ | MDR _S. aureus_ | MDR _E. faecalis_ |
| SCopW | NH₂ | S | L | L | W | I | A | L | R | K | K | CONH₂ | +4 | 1226.6 | >20 | >20 | >20 | >20 |
| HCopW | NH₂ | H | L | L | W | I | A | L | R | K | K | CONH₂ | +5 | 1276.6 | >20 | >20 | >20 | >20 |
| WCopW | NH₂ | W | L | L | W | I | A | L | R | K | K | CONH₂ | +4 | 1325.7 | 20 | 5 | 5 | 10 |
| WCopW1 | NH₂ | W | L | L | W | I | A | L | R | K | K | R | CONH₂ | +5 | 1481.9 | >20 | 5 | **2.5** | 5 |
| WCopW3 | NH₂ | w | l | l | w | i | a | l | r | k | k | r | CONH₂ | +5 | 1481.9 | 5 | **2.5** | **1.25** | 2.5 |
| WCopW5 | NH₂ | w | l | l | w | i | g | l | r | k | k | r | CONH₂ | +5 | 1467.9 | 10 | **1.25** | **1.25** | 2.5 |
| WCopWK | NH₂ | w | l | k | w | i | g | l | r | k | k | r | CONH₂ | +6 | 1482.9 | >20 | >20 | >20 | >20 |
| WCopWE | NH₂ | w | l | e | w | i | g | l | r | k | k | r | CONH₂ | +4 | 1483.8 | >20 | 10 | >20 | >20 |
| WCopWY | NH₂ | w | l | y | w | i | g | l | r | k | k | r | CONH₂ | +5 | 1517.9 | 20 | **1.25** | 10 | 2.5 |
| WCopWV | NH₂ | w | l | v | w | i | g | l | r | k | k | r | CONH₂ | +5 | 1454.8 | 10 | **1.25** | 2.5 | 2.5 |
| WCopWF | NH₂ | w | l | f | w | i | g | l | r | k | k | r | CONH₂ | +5 | 1502.9 | 10 | **1.25** | 2.5 | 2.5 |
| Melittin | | | | | | | | | | | | | | 2846.5 | >20 | **1.25** | 1.25 | 1.25 |

[a]The sequence of the coprisin orthologous α-helical region-derived short AMP, WCopW, and its derivatives. Lower case letters indicate D-form amino acids; capital letters, L-form. Hydrophilic position-preferring residues are remarked as bolded letters because of their critical role in matching the preferred membrane position profile.
[b]The minimal inhibitory concentrations (MICs) were determined in cation-adjusted Mueller–Hinton broth containing 10 mg/l Mg²⁺ and 50 mg/l Ca²⁺ under CLSI conditions. When considering the molecular weight of AMPs (≤1.52 kDa), a MIC of 2.5 μM (≤3.8 μg/ml) (bold letters) could be considered as the "breakpoint" MIC (i.e., <4 μg/ml), but a MIC of 1.25 μM (≤1.9 μg/ml) (bold and underlined letters) is a more confident "breakpoint" MIC (10). Melittin was used as a control because it is one of the most powerful nonclinical AMPs (Table 3 and Supplementary Table S1).

## Results

**Correlation between WCopW5, protegrin-1, and AcWL5 peptide sequences and antimicrobial activity.** The peptide WCopW5 is a derivative of the coprisin orthologous α-helical region. We found that WCopW5 (Table 1) had potent antimicrobial activity and moderate hemolytic activity (Table 1 and Supplementary Fig. 3). The WCopW5 sequence is unlike that of canonical α-helical amphipathic peptides (Fig. 1a), but it resembles the sequence of the C-terminus of protegrin-1 and primary amphipathic AMPs (Fig. 1b). The C-terminal β-strand of protegrin-1 is crucial for the peptide's antimicrobial activity, as previously demonstrated using structural[15,35] and computational[24,36] biology methods and site-directed mutagenesis[25,37,38]. However, the contributions of the Cys residues in the C-terminal β-strand of protegrin-1 to its activity are unclear[16,17,26–28]. The similarity between the Cys-substituted[34] C-terminal β-strand of protegrin-1 and WCopW5 led us to speculate that these two peptides were related. Additionally, the introduction of a hydrophilic residue into the hydrophobic center of WCopWK or WCopWE diminishes the WCopW antimicrobial activity, which is another similarity to protegrin-1[16] (Table 1).

A critical feature distinguishing WCopW5 from the C-terminal β-strand of protegrin-1 is its conformation. To assume a primary amphipathic conformation like that of the protegrin-1 C-terminus, WCopW5 must extend perpendicularly and linearly across the membrane without a stabilizer (such as a Cys residue or a loop). To our knowledge, however, the only peptide that can span a membrane in this manner is AcWL5, the amino acid sequence of which is perfectly matched for positioning within the membrane[30–32] (Fig. 1b). Unexpectedly, an AcWL5 derivative with better membrane-binding activity, WLLWLL, was more similar to WCopW5[30].

We therefore tested whether WCopW5 had substantial antimicrobial activity, given that its amino acid sequence is similar to those of protegrin and AcWL5, in which the amino acids matched the preferred membrane position profile. The antimicrobial activities of derivatives of WCopW5, protegrin, and AcWL5 confirmed the importance of matching the preferred position profile (Table 2). For a peptide to extend perpendicularly and linearly across a membrane without a stabilizer, the conformation must be stabilized by balancing the location of an aromatic ring amino acid and a cationic amino acid on both sides of the hydrophobic carbon chain amino acids (AcWL-1 (WLLLLLLRRR) and AcWL-3 (WLLLLLWRRR)), not just on one

side (AcWL-2 (LLLLLLWRRR)). The absence of this balanced location explains the weak activity of truncated protegrin analogs (Ptg C-ter 1, 2, 3)[37,38]. However, AcWL-1 and AcWL-3 exhibited hemolytic activity and were insoluble, presumably due to their high content of hydrophobic leucine[3] (Table 2 and Supplementary Table 4). On the other hand, the hybrids Ptg C-ter, AcWL-3, WCopW5, Hybrid-3, and Hybrid-4 (LWCopW29, WCopW29) exhibited breakpoint MICs and moderate hemolytic activity (Table 2). The placement of Trp3 in the mid-membrane part of an AMP may be preferred by an adjacent interfacial slab, as observed in the toroidal-pore AMP model (Supplementary Fig. 1).

**Correlation between matching the amino acid preferred membrane position profile and antimicrobial activity.** There are no better AMPs among the secondary amphipathic α-helix AMPs with similar amino acid compositions (LKWLKWLK[39], WLLKRWKKLL[40], RLWLAIKRR[41], KFKWWRMLI[42], KWIKWIKWI[43], AVWKFVKRV[44], KLWWMIRRW[45], and KIWVIRWR[46,47]) (Supplementary Table 5) than WCopW29 and LWCopW29. However, the MICs of AMPs measured under different experimental conditions using different bacterial strains must be compared with caution[41,42,46]. To compare the structures and antimicrobial activities of AMPs with amino acid sequences that match the preferred membrane position profile to those with without matching sequences under the same experimental conditions, we prepared the permutated secondary amphipathic analogs, HLWCopW29-1, 2, 3, and 4 (Supplementary Fig. 2). Using circular dichroism, we observed that these permutated analogs (especially HLWCopW29-2 and HLWCopW29-4 (Supplementary Fig. 4)) had α-helical structures but less antimicrobial activity than LWCopW29 and WcopW29 (Table 2).

Whether amino acid sequences match the preferred membrane positioning profile is determined by measuring the free energy[10]. AMPs with matching amino acid sequences should have a stronger affinity for artificial bacterial membranes (binding constant, K; 1/binding affinity, Kd) and release more free energy ($\Delta G = \Delta H - T\Delta S$) than unmatched analogs. The thermodynamics indicated that WCopW29 and LWCopW29 had a stronger affinity for the membrane than HLWCopW29-2 or HLWCopW29-4 (Fig. 2a and Supplementary Fig. 5). However, thermodynamic parameters provide limited information, as they

**Table 2 Sequence and activity of WCopW5 derivatives.**

| Compound | Sequence[a] | Net charge | MW | Minimum inhibitory concentrations (μM)[b] | | | | Hemolysis percent (%)[c] | | |
|---|---|---|---|---|---|---|---|---|---|---|
| | | | | MDR P. aeruginosa | MDR A. baumannii | MDR S. aureus | MDR E. faecalis | 100 μM | 50 μM | 25 μM |
| Protegrin-1 | NH2–R G G R L C Y C R R–CONH2 / R G R R | +7 | 2155.7 | 10 | 10 | 10 | 10 | 71.66 ±12.24 | 56.75 ±6.02 | 47.83 ±6.23 |
| Ptg C-ter 1 | G G V V V L F R R R–CONH2 | +4 | 1114.4 | 1.25* | 1.25* | 5* | 1.25* | 1.17 ±0.07 | 0.99 ±0.26 | 0.40 ±0.10 |
| Ptg C-ter 2 | G G V L W V L F R R R–CONH2 | +4 | 1187.5 | >20 | >20 | >20 | >20 | 1.80 ±0.07 | 1.14 ±0.02 | 0.52 ±0.05 |
| Ptg C-ter 3 | G G V V W V L F R R R–CONH2 | +4 | 1187.5 | >20 | >20 | >20 | >20 | 1.32 ±0.09 | 0.80 ±0.09 | 0.44 ±0.10 |
| AcWL-1 | W L W L L L L W R R R–CONH2 | +4 | 1237.6 | >20 | 2.5 | 10 | 10 | 36.35 ±3.00 | 20.52 ±1.25 | 11.89 ±1.24 |
| AcWL-2 | L L W L L L L W R R R–CONH2 | +4 | 1237.6 | >20 | >20 | 10 | 10 | 27.97 ±2.79 | 15.13 ± 2.05 | 14.39 ±0.70 |
| AcWL-3 | W L W L W V W W R R R–CONH2 | +4 | 1310.7 | >20 | 2.5 | 5 | 5 | 36.71 ±3.91 | 19.80 ±0.76 | 16.44 ±0.89 |
| Hybrid-1 | W L W L W V G R R R–CONH2 | +4 | 1226.5 | >20 | >20 | >20 | >20 | 1.40 ±0.08 | 1.09 ±0.32 | 0.88 ±0.24 |
| Hybrid-2 | W L W L W I G R R R–CONH2 | +4 | 1240.5 | >20 | 10 | >20 | >20 | 2.88 ±1.49 | 1.21 ±0.01 | 0.66 ±0.07 |
| Hybrid-3 | W L W L W V W R R R–CONH2 | +4 | 1355.7 | >20 | 2.5 | 10 | 10 | 7.17 ±1.09 | 5.14 ±0.19 | 4.26 ±0.63 |
| LWCopW29 (Hybrid-4) | W L W L W I W R R R–CONH2 | +4 | 1369.7 | >20 | 1.25 | 5 | 5 | 13.65 ±0.74 | 5.15 ±0.15 | 0.84 ±0.06 |
| WCopW29 | w l w v w w r r r–CONH2 | +4 | 1369.7 | 10* / 5 / 5* | 0.31* / 0.63 / 0.15* | 1.25* / 1.25 / 0.31* | 1.25* / 1.25 / 0.31* | 12.00 ±1.27 | 5.40 ±0.56 | 0.43 ±0.01 |
| HLWCopW29-1 | R W L R W R W I R V–CONH2 | +4 | 1369.7 | 20 | 2.5 | 10 | 10 | 16.39 ±2.19 | 1.35 ±0.11 | 1.20 ±0.14 |
| HLWCopW29-2 | W L R R W W I R V–CONH2 | +4 | 1369.7 | >20 | 5 | 10 | 10 | 19.60 ±0.50 | 4.13 ±0.28 | 1.87 ±0.12 |
| HLWCopW29-3 | W R L W R W I V R–CONH2 | +4 | 1369.7 | >20 | 20 | 20 | 20 | 4.45 ±0.23 | 1.37 ±0.02 | 1.10 ±0.14 |
| HLWCopW29-4 | R L W W W I V R–CONH2 | +4 | 1369.7 | >20 | 20 | 20 | 20 | 2.47 ±0.03 | 1.08 ±0.04 | 1.02 ±0.01 |

[a]Sequence of protegrin-1 and its C-terminal-truncated AcWL5 and WCopW5 derivatives. Lower case letters indicate D-form amino acids; capital letters, L-form. Hydrophilic position-preferring residues are remarked as bolded letters because of their critical role in matching the preferred membrane position profile.

[b]MICs in the high-salt CLSI standard condition. When considering the molecular weight of AMPs (≤1.52 kDa), a MIC of 2.5 μM (≤3.8 μg/ml), a MIC of 1.25 μM (≤1.9 μg/ml) remarked as bold and underlined letters and a MIC of 1.25 μM (≤1.9 μg/ml) remarked as bold letters. The asterisk (*) indicates the MIC value measured in the presence of albumin (0.2%) and acetic acid (0.01%).

[c]Hemolytic activity measured using 8% human red blood cells in PBS.

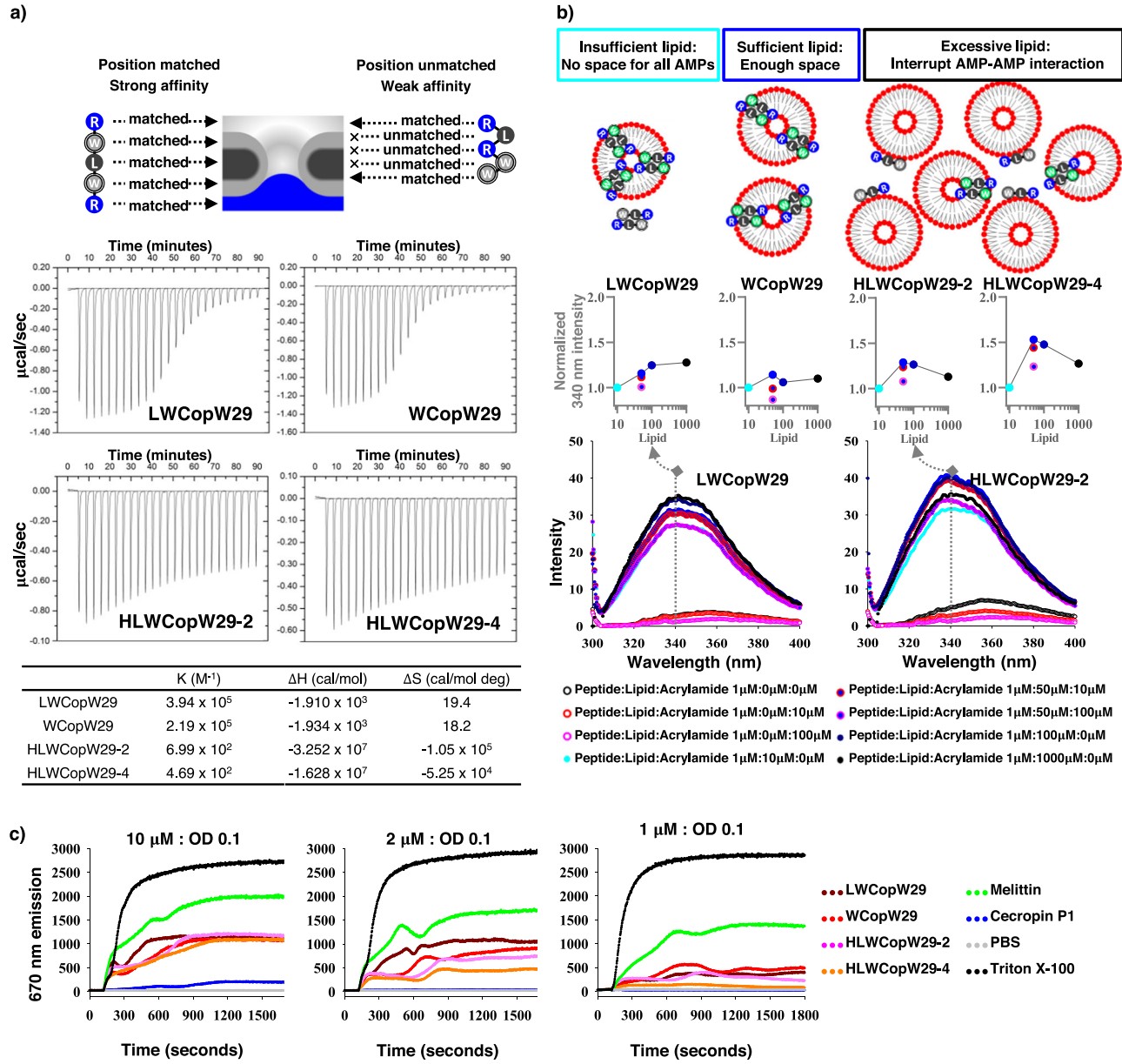

**Fig. 2 Comparisons of peptide-lipid interactions. a** Isothermal titration calorimetry measurements of peptide interactions with DMPC:DMPG liposomes. ΔH indicates the enthalpy change. ΔS indicates the entropy change. The binding constant (or the association constant K, M−1) is the inverse of the dissociation constant (Kd). The larger binding constants of LWCopW29 and WCopW29 indicate these peptides have high affinity for the artificial bacterial membrane liposomes. An independent experiment yielded the same results (Supplementary Fig. 5). **b** Addition of DMPC:DMPG liposomes changed the tryptophan fluorescence intensity of the peptides. In the graphs, the 340 nm maxima intensity values were normalized to the initial fluorescence value to allow comparisons of fluorescence intensities. A quencher (water or acrylamide) decreased the fluorescence intensity, while environmental hydrophobicity (lipid) increased the fluorescence intensity. Lipid concentrations are as follows: light blue, 10 μM; blue, 50 μM; dark blue, 100 μM; black, 1000 μM. The red dotted line indicates 10 μM acrylamide; the pink dotted line, 100 μM acrylamide. A high lipid concentration (1000 μM) quenched HLWCopW29-2 and HLWCopW29-4 fluorescence. This effect was comparable to that of acrylamide, indicting larger exposure of tryptophan to water. Independent experiments yielded the same results (Supplementary Fig. 6). **c** Proton-leakage increases DiSC3(5) fluorescence in *S. aureus*. Peptides were added at 120 s. The fluorescence intensities of the four analogs were similar at the HLWCopW29-2 and HLWCopW29-4 MIC against OD 0.1 *S. aureus* (10 μM peptide each). The fluorescence intensities of LWCopW29 and WCopW29 were higher at the LWCopW29 and WCopW29 MICs (2 μM and 1 μM, respectively) (Supplementary Table 2). An independent experiment yielded the same results (Supplementary Fig. 7).

cannot distinguish between peptide-membrane binding and peptide insertion into the membrane. Tryptophan fluorescence measurements can provide information about peptide insertion into the membrane. A more intense fluorescence signal indicates peptide insertion into the membrane because the hydrophobic environment of the membrane center prevents fluorescence quenching by the water[48–50]. The fluorescence of all analogs was increased by increasing the lipid concentration from a peptide:-lipid ratio of 1 μM:0 μM to 1 μM:100 μM. However, a peptide:-lipid ratio of 1 μM:1000 μM quenched the fluorescence of HLWCopW29-2 and HLWCopW29-4, but not that of LWCopW29 and WCopW29 (Fig. 2b and Supplementary Fig. 6). High lipid concentrations may interfere with AMP–AMP interactions, which are important for membrane insertion (see

AMP–pore ensemble model[5]). The tryptophan fluorescence data thus indicate that WCopW29 and LWCopW29 insert into membranes more efficiently than HLWCopW29-2 or HLWCopW29-4 (Fig. 2b and Supplementary Fig. 6). This finding, obtained using artificial bacterial membranes, corresponds to the antimicrobial activities of WCopW29. If WCopW29 and LWCopW29 efficiently insert into an excessive concentration of artificial bacterial membrane, they would also be expected to insert into an excessive concentration of true bacteria membrane. When the *S. aureus* concentration was increased 200-fold, LWCopW29 and WCopW29 MIC were increased only twofold, while HLWCopW29-2 and melittin MIC were increased 8-16-fold (Supplementary Fig. 10 and Supplementary Table 2). Transmembrane potential assays revealed that membrane permeation was induced by AMP insertion. LWCopW29 and WCopW29 induced greater permeation of the bacterial membrane than HLWCopW29-2 or HLW-CopW29-4 (Fig. 2c and Supplementary Fig. 7). AMPs induce membrane permeation through poration or micellization[5,6]. Stable, uniform and constant-sized liposome volumes indicate poration, while dynamic and heterogeneous liposome volumes indicate micellization[51]. We found that both preferred position-matched and -unmatched AMPs permeated membranes via poration[52] (Supplementary Figs. 8 and 9). We also observed that LWCopW29 interacted more strongly with membranes than WCopW29, which has stronger antimicrobial activity. These results demonstrate the importance of the protease resistance conferred by D-amino acids[33,53–55] (Supplementary Fig. 12).

In summary, our findings indicate that preferred position-matched AMPs insert into membranes more efficiently than unmatched AMPs, which improves both membrane permeation and antimicrobial activity.

### Conserved advantageous properties of antimicrobial peptides.
WCopW29 exhibits better membrane binding, insertion and permeation as well as a better antimicrobial activity than canonical helical counterparts. However, it is too early to conclude that its antimicrobial activity is caused its membrane binding, insertion, and permeation. For example, murepavidin[22] and peptide 3[56], which are two other protegrin-inspired AMPs, acquire better antimicrobial activity from a stereospecific, porin-targeting mechanism, but that results in a loss of AMP competitive advantage with respect to its broad spectrum antimicrobial activity and stability against drug resistance[22,56]. For WCopW29, therefore, it is necessary to beware of a similar loss of the AMP advantage when considering its similarity to protegrin-inspired antibiotics.

The broad spectrum membrane damaging activity of AMPs can be estimated by measuring nitrocefin and ONPG degradation. Nitrocefin degradation is a sign of outer membrane damage, while ONPG degradation is a sign of inner membrane damage. Using these criteria, we found that WCopW29 damages both the outer and inner membrane (Fig. 3a). Electron micrographs obtained using field emission scanning electron microscopy (FE-SEM) showed the time-dependent increase in damage to the outer and inner membranes (Fig. 3b). The bactericidal activity of WCopW29 was further confirmed by its ability to reduce the numbers of colony-forming units (CFUs) (Fig. 3c). The stability of AMPs against the acquisition of drug resistance by bacteria was assessed by treating bacteria with sub-MIC concentrations of the AMPs (Fig. 3d). We found that the MIC of WCopW29 was stable over a period of 30 days, indicating that WCopW29 suppresses the emergence of WCopW29-resistant in bacteria. Collectively, these results indicate that WCopW29 possesses many of the advantages of AMPs. The better antimicrobial activity of

WCopW29 was the result of membrane permeation facilitated by matching its amino acid sequence to that preferred for membrane insertion.

### Generation of next-generation antimicrobial peptides by matching amino acid preferred membrane position profiles.
The results summarized above suggest that we could potentially generate AMPs possessing a degree of antimicrobial activity that has only rarely been achieved before. We therefore used preferred position profiling to design and test a series of WCopW29 derivatives (Table 3 and Supplementary Table 1). Most derivatives had breakpoint MICs (at least two MICs ≤1.25), and their sequences strongly matched those preferred for membrane insertion. We found that inserting hydrophilic amino acids (Lys, Asp) that matched those for positioning in the membrane center increased the MIC (WCopW37, 38), but inserting them such that they matched those positioned for the membrane surface did not (WCopW35, 36). The surface-matched position even tolerated the addition of glucose (WCopW47, 48).

WCopW43 exhibited in vitro activity comparable to the activities of colistin and daptomycin; that is, it showed moderate hemolytic and bactericidal activities (Table 3, Fig. 4a–c, and Supplementary Fig. 13). Because WCopW43 showed MIC breakpoint activity against colistin-resistant *A. baumannii*, we tested it in vivo antimicrobial activity in BALB/c mice subcutaneously infected with MDR *A. baumannii*, MDR *S. aureus* or MDR *K. pneumonia* (Supplementary Table 6) (Fig. 4d–f). The results confirmed the in vivo efficacy of WCopW43. Following MDR *A. baumannii*, MDR *S. aureus* or MDR *K. pneumonia* infection, (Fig. 4e), mice injected with non-nephrotoxic concentrations of WCopW43 (Fig. 4d and Supplementary Fig. 14) survived longer than those injected with the phosphate-buffered saline (PBS). WCopW43 injection also dramatically decreased the bacterial burden by ~90% (Fig. 4f).

### Discussion
In this study, we developed colistin- and daptomycin-comparable AMPs through rational design. Thanks to advances in screening methods, a number of potent AMPs have been acquired through cost-effective and creative random screening[3,54]. On the other hand, AMPs rationally designed based on an advanced understanding of AMPs were less effective than AMPs generated through repetitive trial-and-error[19,57]. This is confusing, given that the structures of the rationally designed AMPs were based on an extensive understanding of the complex membrane–peptide interactions. Nevertheless, we believe that rational understanding can provide guidance that cannot be obtained in other ways.

Occasionally, we obtained unique AMPs like the WCopWs, the activity of which cannot be explained by the canonical structure–activity relationship of AMPs. However, extensive and detailed comparison of the structure–activity relationships of sequentially similar peptides, membrane–peptide interaction data, and the site-directed modified analogs guided us to optimize the sequences of AMPs by matching the amino acid preferred position profile of the membrane. This profile-matching guided us to rationally design WCopW43, which is more active than any previously described nonclinical AMPs.

This rational approach also guides further study. For example, we could try to conjugate vancomycin or colistin to surface position-preferring residues to produce a synergistic effect[58] without disrupting peptide-membrane insertion. Profile matching can be more perfectly optimized by modulating backbone length with various peptidomimetics like β-amino acids[4]. If an accurate amino acid preferred position profile for lipopolysaccharide or cholesterol was obtained, a more membrane-selective candidate

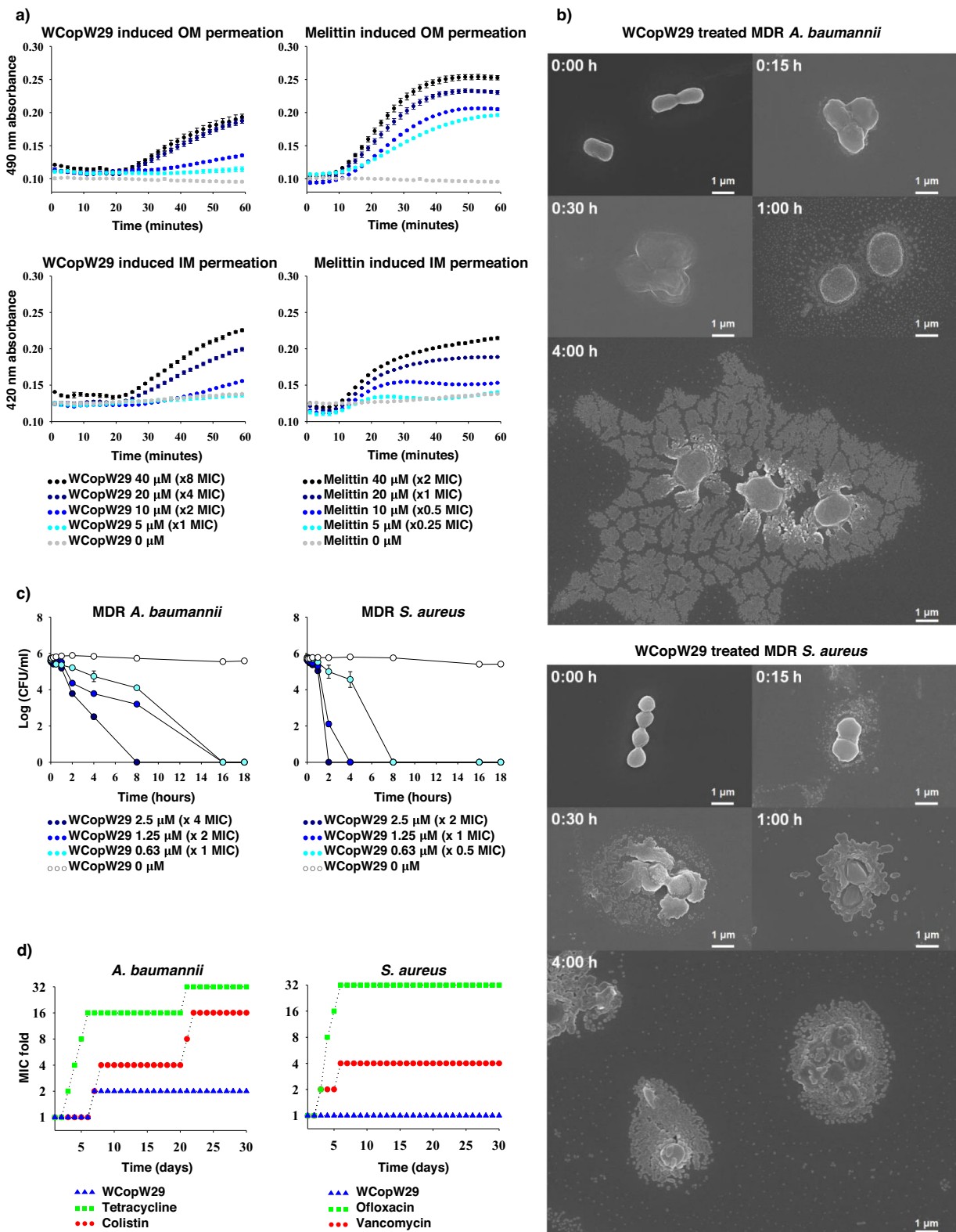

a) WCopW29 induced OM permeation / Melittin induced OM permeation / WCopW29 induced IM permeation / Melittin induced IM permeation

•••WCopW29 40 µM (x8 MIC)
•••WCopW29 20 µM (x4 MIC)
•••WCopW29 10 µM (x2 MIC)
•••WCopW29 5 µM (x1 MIC)
•••WCopW29 0 µM

•••Melittin 40 µM (x2 MIC)
•••Melittin 20 µM (x1 MIC)
•••Melittin 10 µM (x0.5 MIC)
•••Melittin 5 µM (x0.25 MIC)
•••Melittin 0 µM

b) WCopW29 treated MDR *A. baumannii*
WCopW29 treated MDR *S. aureus*

c) MDR *A. baumannii* / MDR *S. aureus*

•••WCopW29 2.5 µM (x 4 MIC)
•••WCopW29 1.25 µM (x 2 MIC)
•••WCopW29 0.63 µM (x 1 MIC)
ooo WCopW29 0 µM

•••WCopW29 2.5 µM (x 2 MIC)
•••WCopW29 1.25 µM (x 1 MIC)
•••WCopW29 0.63 µM (x 0.5 MIC)
ooo WCopW29 0 µM

d) *A. baumannii* / *S. aureus*

▲▲▲ WCopW29
■■■ Tetracycline
••• Colistin

▲▲▲ WCopW29
■■■ Ofloxacin
••• Vancomycin

could be designed. A preferred position profile could also be applied to the development of cell-penetrating peptides (CPPs). Although the similarity between AMPs and CPPs is well known, profile-matched peptides are more closely correlated with CPPs when considering that primary amphipathy is the major common property of CPPs. With a simple idea and few technical requirements, preferred position profile-matched peptides have the potential for extensive applicability.

Although it was not a focus of this study, β-strand conformation is also important for the activity of WCopWs (Supplementary Fig. 1). Strong activity correlated with the appearance of a negative 218 nm circular dichroism peak (Supplementary Fig. 4). It is known that an extended β-strand conformation is

**Fig. 3 Antimicrobial peptides advantageous properties of WCopW29. a** Time- and concentration-dependent permeation of the outer and inner membrane (probed with nitrocefin and ONPG degradation, respectively) indicate that WCopW29 permeates both membranes at MIC concentrations. **b** Scanning electron micrographs showing the time-dependent swelling and peeling off of the membranes of MDR Gram-negative and Gram-positive bacteria exposed to MIC concentrations of WCopW29 over a period of 4 h (Supplementary Figs. 10 and 11). **c** Antimicrobial kinetics of MDR Gram-negative and Gram-positive bacteria indicate that WCopW29 kills both at MIC concentrations. **d** Suppression of resistance acquisition by non-resistant Gram-negative and Gram-positive bacteria exposed to MIC and sub-MIC concentrations of WCopW29 for 30 days. Stable antibiotics, which mostly target rarely mutable targets (e.g., colistin for lipopolysaccharide and vancomycin for peptidoglycan pentapeptide), and stereospecific antibiotics, which target readily mutable proteins (e.g., tetracycline for ribosomes and ofloxacin for DNA gyrase) were used as controls.

---

essential for construction of a pore structure by 9- or 10-mer peptides[11,35,39,59]. Loss of activity caused by introducing L- and D-amino acid repeats into WCopW65, thereby disrupting its secondary structure, also demonstrate the importance of the β-strand conformation[60]. β-strand may be as important as preferred position profile matching because the interstrand hydrogen bonding of β-strands could provide a stabilizing force for the linear structure from the side, while profile matching could provide a stabilizing force at both terminals.

Earlier studies have suggested that β-strand AMPs have advantages over α-helical AMPs. Phospholipids do not have a straight, cylinder geometry; instead, differences in head and tail diameters lead to a conical geometry[6,13]. The "wedge" or "void space" between cones could facilitate the insertion of AMPs into the membrane core by bypassing hydrophobic–hydrophilic repulsion and lateral pressure. Theoretically, therefore, AMPs whose amino acid sequences are designed with β-strand to optimize membrane insertion are more advantageous than α-helical AMPs because the thin β-strand inserts more easily into the void space[13] (Supplementary Fig. 1).

Longer length with fewer residues is another advantage that β-strand AMPs have over α-helical peptides. Membrane spanning length is important for α-helical amphipathic AMP activity. However, α-helical amphipathic AMPs are limited by their high molecular weight[4], which cannot be avoided because longer sequences are needed for the stable formation of α-helical amphipathic structures that span the membrane. If the molar concentration is taken into consideration, the antimicrobial activities of the most active α-helical amphipathic AMPs (such as melittin) are comparable to that of WCopW5 (Supplementary Table 1). However, melittin (2.85 kDa) is two times larger than WCopW5 (1.47 kDa), so its actual MIC (3.56 μg/ml) is half that of WCopW5 (1.84 μg/ml), as antimicrobial activity is not expressed in terms of molar concentration but as weight/volume concentration[8].

However, the key feature of β-strand AMPs over α-helical AMPs is clearly their compatibility with the preferred position profile. It is noteworthy that regardless of how we optimize α-helix peptides to the preferred position profile, its most stable state is a surface-binding state. For most α-helical AMPs, the transmembrane state is merely an intermediate that occurs when translocating from an outer leaflet surface-bound state to an inner leaflet surface-bound state[5,61]. This is the reason that most α-helical AMP-induced pores (except a few true poration AMPs, such as melittin[61], or β-helical AMPs, such as gramicidin A[62]) are prone to be transient. By contrast, by matching their sequence to the preferred position profile, β-strand AMPs can be stabilized in a transmembrane state. Although we lack direct observation data, we expect that this stable transmembrane state would be the most advantageous property of LWCopW29 and WCopW29 over HWCopW29-1,2,3,4.

Our profile-matching AMP design has some critical limitations. Although we designed the AMP sequences to match the preferred position profile, we did not observe the allocation in a real membrane, so we cannot be certain that the hydrophobic position-preferring residues (WIWVLW) were actually situated within the hydrophobic core and that the hydrophilic position-preferring residues (NH2, CONH2, RRR, K) were actually situated at the surface. Instead, we deduced their location from two pieces of indirect evidence. First, the conserved MIC after glycosylation of WCopW47, 48 indicates that neither terminal was a membrane-binding residue[63]. The only remaining candidates for the membrane-binding residues are the hydrophobic position-preferring residues.

Second, the peptide-membrane interaction data (Fig. 2 and Supplementary Figs. 8 and 9) indicates that WCopWs induce membrane poration[61]. There are two membrane poration models: the transmembrane model and the interfacial activity model. In the transmembrane model, hydrophobic position-preferring residues are located within a hydrophobic core. In the interfacial activity model, all residues of the peptide are evenly located around an interface[60,61]. Thus, if WCopWs do not correspond to the interfacial activity model, it can be concluded that the hydrophobic position-preferring resides are indeed located within the hydrophobic core. In the interface activity model, changing the hydrophobic position-preferring residues from L-amino acids to D-amino acids would not induce a clear change in activity[60]. By contrast, with our WCopWs, D- to L-amino acid substitution of hydrophobic position-preferring residues (WCopW65) eliminated the peptide's activity. In addition, single D- to L-amino acid substitution of hydrophilic position-preferring residues did not induce a clear change in the activities of WCopW55, 62, 63, 64. This suggests that the location of the hydrophobic and hydrophilic position-preferring residues are not even. The only existing model consistent with those observations without conflict is the transmembrane poration model. (If we had performed the tryptophan quenching experiment with WCopW47, 48, and WCopW65 as well as with the glucosylated hydrophobic position-preferring residue analog or the single D- to L-amino acid substituted hydrophobic position-preferring residue analog, which exhibited a greatly deteriorated MIC, our evidence would have been reinforced, but, unfortunately, we also did not do that experiment.)

We also want to discuss the "missing links" in what we know about the mechanism. We observed and compared the binding, insertion, permeation, and activity of the tested AMPs. However, we lack data on the events between insertion and permeation and between permeation and activity. To link insertion and permeation, we will need greater insight into the AMP-lipid interaction, including pore structure and pore life span. Such information could be gained through the use of molecular dynamics simulations or nuclear magnetic resonance analyses. Without those data, for now, we must make inferences based on our observation as well as earlier reported work. One interesting observation was that permeation by LWCopW29 was similar to that of HLWCopW29 when the peptide:lipid ratio was high. This is most likely because at high peptide:lipid ratios surface-binding AMPs (HLWCopW29-2,4) can induce permeation as effectively as transmembrane pore-inducing AMPs (LWCopW29, WCopW29)[61]. We also suspect that micellization of LWCopW29

**Table 3 Sequence and activity of WCopW29 derivatives.**

| Compound | Sequence[a] | Net charge | MW | Minimum inhibitory concentrations (μM)[b] | | | | Hemolysis percent (%)[c] | | |
|---|---|---|---|---|---|---|---|---|---|---|
| | | | | MDR P. aeruginosa | MDR A. baumannii | MDR S. aureus | MDR E. faecalis | 100 μM | 50 μM | 25 μM |
| WCopW44 | NH₂ … CONH₂ | +4 | 1254.5 | 5 | **2.5** | **1.25** | **1.25** | 4.48 ±1.01 | 0.87 ±0.36 | 0.11 ±0.03 |
| WCopW41 | NH₂ … CONH₂ | +4 | 1270.6 | >5 | **2.5** | **2.5** | **1.25** | 0.42 ±0.06 | 0.04 ± 0.02 | 0.01 ±0.01 |
| WCopW32 | NH₂ … CONH₂ | +3 | 1213.5 | >5 | **1.25** | **1.25** | **1.25** | 4.53 ±0.01 | 2.95 ± 0.21 | 1.66 ± 0.11 |
| WCopW29 | NH₂ … CONH₂ | +4 | 1369.7 | 5 | **0.63** | **1.25** | **1.25** | 10.04 ± 0.88 | 6.96 ± 0.13 | 2.18 ± 0.16 |
| WCopW35 | NH₂ … CONH₂ | +5 | 1497.8 | **2.5** | **0.63** | **0.63** | **0.63** | 6.07 ±1.01 | 2.71 ±0.63 | 1.18 ±0.34 |
| WCopW36 | NH₂ … CONH₂ | +3 | 1484.8 | >5 | 5 | **1.25** | **1.25** | 9.86 ±1.62 | 7.48 ±0.90 | 4.03 ±0.40 |
| WCopW37 | NH₂ … CONH₂ | +5 | 1384.7 | >5 | >5 | 5 | 5 | 0.16 ±0.03 | 0.08 ±0.01 | 0.13 ±0.02 |
| WCopW38 | NH₂ … CONH₂ | +3 | 1371.6 | >5 | **2.5** | >5 | >5 | 0.06 ± 0.03 | 0.06 ±0.06 | 0.06 ±0.06 |
| WCopW61 | NH₂ … CONH₂ | +4 | 1383.7 | 5 | **1.25** | **0.63** | **0.63** | 11.6 ±0.99 | 8.91 ± 0.75 | 6.90 ± 0.03 |
| WCopW60 | NH₂ … CONH₂ | +4 | 1383.7 | 5 | **1.25** | **0.63** | **0.63** | 15.17 ± 0.19 | 15.14 ± 3.22 | 15.51 ± 0.64 |
| WCopW54 | NH₂ … CONH₂ | +4 | 1369.7 | **2.5** | **0.63** | **0.63** | **0.63** | 35.13 ± 4.13 | 11.43 ± 2.42 | 8.38 ± 1.14 |
| WCopW43 | NH₂ … CONH₂ | +5 | 1497.9 | **1.25** | **0.63** | **0.63** | **0.63** | 5.02 ±0.63 | 2.22 ± 0.67 | 1.12 ± 0.20 |
| WCopW47 | NH₂ … CONH₂ | +5 | 1815.5 | >5 | **1.25** | **2.5** | **2.5** | 2.07 ±0.23 | 1.19 ±0.17 | 0.73 ±0.05 |
| WCopW48 | NH₂ … CONH₂ | +5 | 1815.5 | **2.5** | **0.63** | **2.5** | **2.5** | 0.85 ±0.02 | 0.69 ±0.04 | 0.45 ±0.03 |
| Melittin | | | 2846.5 | >5 | **1.25** | **1.25** | **1.25** | 97.66 ±1.64 | 94.83 ±0.22 | 102.53 ±6.34 |
| Colistin | | | 1155.5 | **0.63** | **0.31** | >5 | >5 | 0.39 ±0.06 | 0.31 ±0.01 | 0.24 ±0.09 |
| Daptomycin | | | 1619.7 | >5 | >5 | **1.25** | **1.25** | 0.04 ±0.02 | 0.03 ±0.01 | 0.01 ±0.02 |

[a]Sequences of WCopW29 derivatives. Lower case letters indicate D-form amino acids; capital letters, L-form. Hydrophobic position-preferring residues are remarked as bolded letters. N of WCopW47 and WCopW48 indicate glucosylated asparagine.
[b]MICs in the high-salt CLSI standard condition. When considering the molecular weight of AMPs (≤1.52 kDa), a MIC of 2.5 μM (≤3.8 μg/ml) remarked as bold letters and a MIC of 1.25 μM (≤1.9 μg/ml) remarked as bold and underlined letters. Colistin (clinical AMP against Gram-negative MDR bacteria), daptomycin (clinical AMP against Gram-positive MDR bacteria), and melittin (one of most potent, nonclinical AMPs) were used as controls (Supplementary Table S1).
[c]Hemolytic activity to 8% human red blood cells in PBS.

or WCopW29 may limit permeation at high concentrations because primary amphipathic peptides tend to micellize themselves at high concentrations[64].

Another interesting observation was that insertion and permeation by LWCopW29 were consistently better than by WCopW29. This is most likely because normal peptides composed of L-amino acids tend to have a stronger affinity for the normal liposomes, while peptides composed of D-amino acids tend to have a stronger affinity for similarly enantiomeric liposomes[53]. Presumably, the better activity of WCopW29 relies solely on protease stability (Supplementary Fig. 12).

Finally, there is a lack of proportionality between permeability and activity. At 10 μM, melittin induces more than 3 times as much permeation as LWCopW29. Despite this lower level of permeation, LWCopW29 completely inhibits bacteria but melittin does not (Supplementary Table 2). The easiest explanation for this result is the dual-targeting of LWCopW29. However, the conserved and broad antibacterial spectrum observed with racemized, reverted, and point-mutated LWCopW29 analogs is sufficient to reject the possibility of dual-targeting. Consequently, at present we cannot explain this inconsistency. Indeed, the lack of proportionality between permeability and activity is the most common and critical obstacle to a rational approach to AMP development. For example, daptomycin does not induce leakage of fluorescent probes, though it is a strong antimicrobial agent[65,66]. There are many examples of AMP derivatives with permeation activities that are disproportionate with respect to their antimicrobial activities[28,57,60]. But there are also many examples where these activities are proportionate[46,49,67].

One possible explanation for the lack of proportionality between permeation and antimicrobial efficacy is negative feedback inhibition of AMPs by leaked cytosolic contents. Macromolecules and salts rapidly released by AMP-induced membrane poration may suppress AMP activity by disrupting electrostatic interactions by binding AMPs or by inducing stress responses[68,69]. If so, considering that AMPs do not need the same level of permeability as melittin because bacterial metabolism can be critically disrupted by partial membrane permeation[70], we would not need to pursue a peptide with marked permeability. Instead, AMPs that can stably induce moderate permeation under challenging conditions (e.g., presence of protease, low peptide:lipid ratio, low residue number, or high-salt concentration) may be pursued as a goal of rational design of potent AMPs.

## Methods

**Bacteria preparation.** MDR *Pseudomonas aeruginosa* (CCARM 2180) was purchased from the Culture Collection of Antimicrobial Resistance Microbes (Seoul, Republic of Korea). MDR *Acinetobacter baumannii* (ATCC BAA-1605) and MDR *Enterococcus faecalis* (ATCC 51575) were purchased from the American Type Culture Collection (Manassas, VA, USA). MDR *Staphylococcus aureus* (KCCM 40510) was purchased from the Korean Culture Center of Microorganisms (Seoul, Republic of Korea). *S. aureus* (KCTC 1621) was purchased from the Korean Collection for Type Cultures. *A. baumannii* (KCCM 40203) was purchased from the Korean Culture Center of Microorganisms (Seoul, Republic of Korea).

To determine MIC50s and MIC90s, 28 *A. baumannii* strains and 10 *Klebsiella pneumoniae* strains were prepared. Twenty-six *A. baumannii* strains were clinically isolated and gifted to us by the Kyungpook National University School of Medicine. One *A. baumannii* strain (ATCC 17978) was purchased and gifted to us by Kyungpook National University School of Medicine. One *A. baumannii* strain and 10 *K. pneumoniae* strains (NCCP) were purchased from the National Culture Collection for Pathogens.

**Antimicrobial peptide preparation.** All L-form WCopW analogs (except YCopW, SCopW, HCopW, WCopW, WhCopW, WtCopW, and WCopW1, 2) were prepared using solid-phase peptide synthesis and validated with HPLC (Shimadzu, Kyoto, Japan) and matrix-assisted laser desorption ionization time-of-flight mass spectrometry (Shimadzu, Kyoto, Japan). Other L-form and D-form WCopW analogs as well as cecropin P1, melittin, and protegrin-1 (linear) were purchased from Anygen Inc. (Gwangju, Republic of Korea). Protegrin-1 was refolded with 2 mM glutathione (Bioworld, Tokyo, Japan) and 0.2 mM glutathione disulfide (Bioworld,

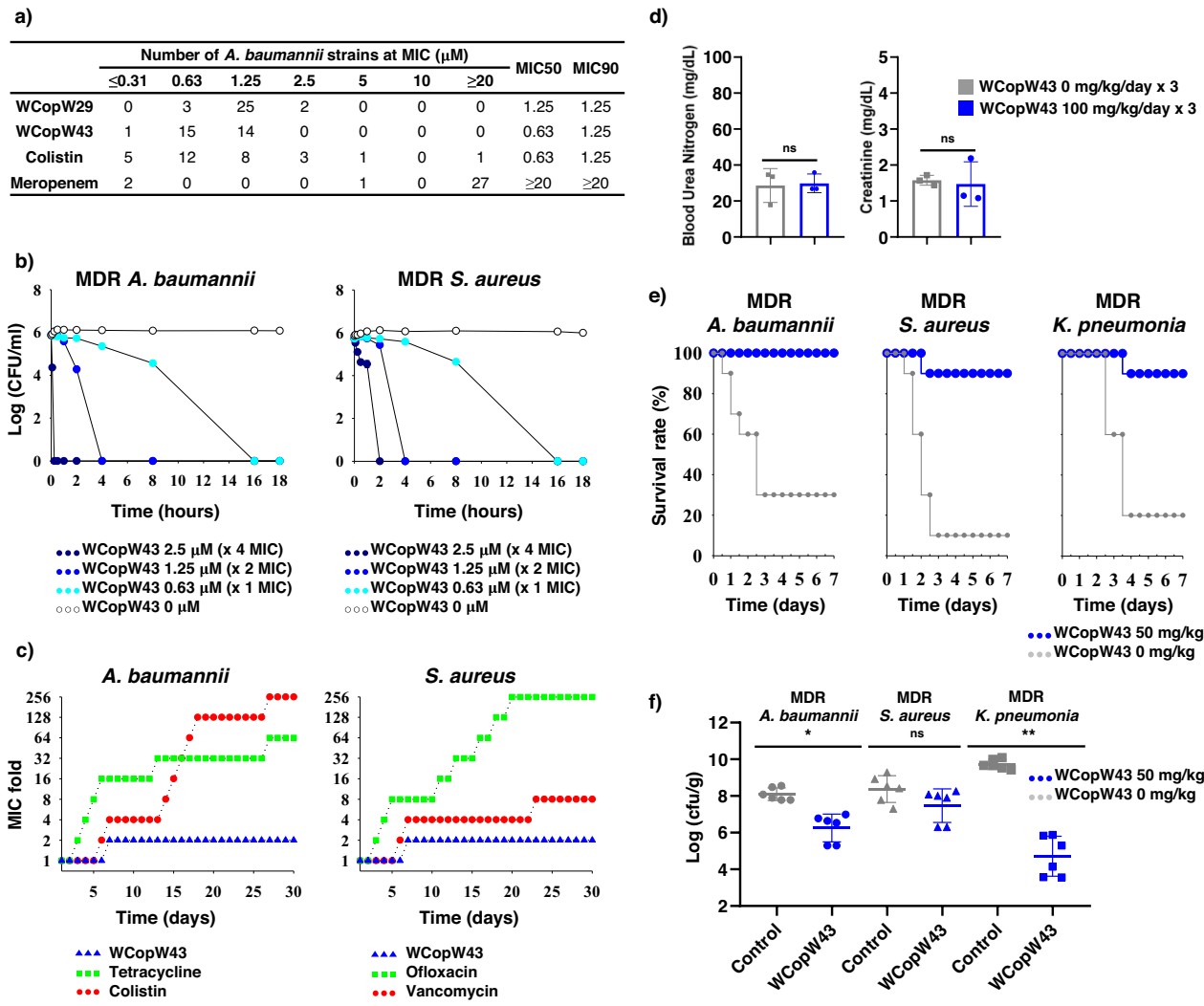

**Fig. 4 WCopW43 is a potential next-generation AMP. a** The MIC50 and MIC90 values for WCopW43 against 30 *A. baumannii* strains, including 27 carbapenem-resistant strains, indicate that it has a broad spectrum breakpoint MIC against MDR strains, including colistin-resistant strains (Supplementary Table 6). **b** The antimicrobial kinetics of MDR Gram-negative and Gram-positive bacteria indicate that WCopW43 kills both types at MIC concentrations. **c** Analysis of non-resistant Gram-negative and Gram-positive bacteria exposed to MIC and sub-MIC concentrations WCopW43 for 30 days indicates emergence of WCopW43 resistance is suppressed. Stable antibiotics and stereospecific antibiotics were used as controls. **d** Nephrotoxicity indicators in a mouse model. WCopW43 (100 mg/kg) or PBS was subcutaneously injected into BAKB/c mice daily for 3 days. Groups were statistically compared using unpaired *t* tests (*$P < 0.05$, **$P < 0.01$, ns: not significant). **e** Survival rates among infected mice. BALB/c mice were subcutaneously infected with MDR *A. baumannii* ($1 \times 10^8$ cfu/ml), MDR *S. aureus* ($1.5 \times 10^8$ cfu/ml) or MDR *K. pneumonia* ($1 \times 10^9$ cfu/ml). One hour later, the mice were subcutaneously administered WCopW43 (50 mg/kg) or PBS. **f** CFU kinetics of the mouse model. From among the mice in the survival experiment, three were randomly selected. One day after infection, tissue was collected, processed and spread to count CFUs. CFU numbers were statistically compared using unpaired *t* tests.

Tokyo, Japan) for 48 h. The purity and molecular weight of each peptide were measured using HPLC (Shimadzu, Kyoto, Japan) and matrix-assisted laser desorption ionization time-of-flight mass spectrometry (Shimadzu, Kyoto, Japan). All WCopW analogs and cecropin P1, melittin and protegrin-1 were prepared as TFA salts except the WCopW43 used in the in vivo test, which was an acetate salt.

**Liposome (LUV) preparation**. Liposomes were prepared using 1,2-dimyristoyl-sn-glycero-3-(phospho-rac-(1-glycerol)) (sodium salt) (DMPG) (Avanti, Alabama, USA); 1,2-dimyristoyl-sn-glycero-3-phosphocholine (DMPC) (Avanti); and cholesterol (Sigma, Darmstadt, Germany) were used. Bacterial membrane mimic DMPC/DMPG (7/3) unilamellar vesicles (LUVs), and mammalian membrane mimic DMPC/cholesterol (10/1) vesicles with 100-nm diameters were prepared using the freeze-thaw and extrusion method. Lipids were mixed and dissolved in methanol in round-bottomed flasks then evaporated and freeze-dried overnight, yielding a thin layer of lipid covering the glass wall. PBS was added to the lipid layer, after which the PBS-lipid layer was vortexed then frozen in liquid nitrogen and thawed in 50 °C water repeatedly until no visible layer remained. LiposoFast

Liposome Factory Basic unit with stabilizer (Avestin, Ottawa, Canada) was then used to extrude the liposomes 50 times through a polycarbonate membrane filter (pore diameter, 100 nm; filter diameter, 0.75 inches) (Avestin). For circular dichroism, isothermal titration calorimetry, tryptophan fluorescence, and dynamic light scattering experiments, liposomes were reconstructed and resuspended in PBS (Welgene Inc, Gyeongsangbuk-do, Korea).

**Determination of minimal inhibitory concentration**. To assess antimicrobial activity, the MIC50 and MIC90 were measured using broth microdilution following CLSI guidelines. The minimal concentrations inhibiting the growth of 50% and 90% of strains were regarded as MIC50 and MIC90 values, respectively. Each experiment was performed in triplicate. Meropenem (Sigma, Darmstadt, Germany) and colistin were used as control antibiotics. Cation-adjusted Mueller–Hinton broth (CAMHB) (Becton, Dickinson and Company, Franklin Lakes, USA) with supplied calcium (50 mg/L) and magnesium (10 mg/L) was used for the CLSI-test condition. Bacteria in midlog phase and peptides were separately diluted in CAMHB. The bacteria and peptide solutions were then mixed in 96-well

polyethylene plates (SPL, Gyeonggi-do, Korea) and incubated for 18 h at 37 °C. The final concentration for bacteria was $5 \times 10^5$ CFU/ml. The final peptide concentrations were 20 μM, 10 μM, 5 μM, 2.5 μM, 1.25 μM, 0.63 μM, and 0.31 μM or 5 μM, 2.5 μM, 1.25 μM, 0.63 μM, 0.31 μM and 0.15 μM. The MIC was defined to be the lowest concentration of peptide without visible turbidity by comparing with control antibiotics with the unaided eye. Colistin and melittin served as controls for Gram-negative bacteria, while daptomycin, vancomycin and melittin were used as controls for Gram-positive bacteria. Each experiment was performed in triplicate. MICs in albumin and acetic acid solutions were measured using the same protocol but with 0.2% bovine serum albumin (Bovogen, Keilor, Australia), 0.01% acetic acid (Merck, New Jersey, USA), and CAMHB with both calcium and magnesium. In addition, MICs used in the antibiotic resistance acquisition test, the outer–inner membrane permeation test, and the transmembrane potential test and for FE-SEM were individually determined. The final bacterial concentrations used were as follows: for resistance acquisition, optical density (OD) 0.01; for outer–inner membrane permeability assays (LB), OD 0.1; for the transmembrane potential test, $1 \times 10^8$ CFU/ml (OD 0.171); for FE-SEM specimen preparation. Inhibitory activity unidentifiable by direct observation was determined by counting the CFUs. Each experiment was performed at least twice.

**Hemolytic activity measurements.** Human red blood cells (Zenbio, North Carolina, USA) and peptides were separately diluted in PBS (blood concentration, 8%). The blood cell and peptide solutions were then mixed in 96-well polyethylene plates (SPL, Gyeonggi-do, Korea) and incubated for 18 h at 37 °C. The final concentration of human red blood cells was 4%, while peptide concentrations were 100 μM, 50 μM, 25 μM, 12.5 μM, 6.25 μM, 3.13 μM, 1.56 μM, and 0.78 μM. After incubation, the plates were centrifuged at $1000 \times g$ for 5 min, and the supernatants were transferred to 96-well polyethylene plates. Hemolysis levels were determined by measuring the absorbance at 540 nm with a SpectraMax plate spectrophotometer (Molecular Devices, California, USA). Control samples were treated with 1% Triton X-100 (for 100% hemolysis) or solution with 0 μM peptide (for 0% hemolysis) and melittin (for positive control). Each experiment was performed in triplicate.

**Tryptophan fluorescence measurements.** PC/PG liposome stock solutions, peptide solutions, and acrylamide solutions were diluted in PBS. The solutions were then mixed to yield the following final concentrations: lipid, 1000 μM, 100 μM, 50 μM, 10 μM or 0 μM; peptide, 1 μM or 0 μM; acrylamide, 100 μM, 10 μM, or 0 μM. Mixtures were incubated for at least 1 h, after which aliquots were transferred into a four-clear-sided quartz cuvette (Hellma, Mulheim, Germany). The cuvette was placed into an LS 55 fluorescence spectrometer (PerkinElmer, Massachusetts, USA), and measurement parameters were as follows: excitation, 295 nm; emission, 300-400 nm; slit 5; speed, 300; accumulation, 10 times. Blank solutions for each condition contained the following: liposomes, 1000 μM, 100 μM, 50 μM, 10 μM or 0 μM; peptide, 0 μM; acrylamide, 100 μM, 10 μM or 0 μM. Each experiment was performed at least twice. To aid visualization, the intensity of the 340-nm peak in the inset was normalized by the fluorescence intensity of the control with the lowest fluorescence signal (peptide, 1 μM; lipid, 10 μM; acrylamide, 0 μM). Detailed information about the AMP conformation, such as the insertion depth and orientation of insertion, could not be obtained because of tryptophan heterogeneity (50) and the differential fluorescence of D-Trp[48].

**Isothermal titration calorimetry (ITC) measurements.** PC/PG liposome stock solutions and peptide solutions were diluted in PBS and degassed for 30 min. Peptide solutions (2 ml, 40 μM) were then loaded into a VP-ITC MicroCalorimeter (MicroCal, Northampton, USA). Liposome samples (600 μl, 2000 μM) were injected into the VP-ITC using a syringe. Measurement parameters were as follows: cell temperature, 37 °C; total injections, 30; reference power, 10; initial delay, 300; stirring speed, 300; edit mode, 10.0/12.0/200/2. Liposome solution (10 μl) was injected into the peptide solution 30 times. Each experiment was performed in duplicate. The results were calculated using MicroCal analysis.

**Bacterial membrane ion permeation measurements.** MDR *S. aureus* (KCCM 40510) was cultured in CAMHB at 37 °C, washed, and diluted with PBS supplemented with 25 mM glucose to achieve an OD 0.111. DiSC3(5) (final concentration, 2 μM) was then added, and the bacterial suspension was incubated for 1 h at room temperature. Suspensions of bacteria and DiSC3(5) solution (90 μl) were transferred into black, clear-bottomed 96-well plates (Costar, New York, USA). AMPs diluted in PBS/glucose were added to 96-well polyethylene plates (150 μl/well). Bacteria-containing and peptide-containing plates were inserted into a Flexstation 3 Multi-Mode Microplate Reader (Molecular Devices, California, USA). Peptide solutions (10 μl each) were automatically transferred and mixed with bacterial suspensions to yield the following final concentrations: bacteria, OD 0.1; peptides, 10 μM, 2 μM, 1 μM, 0.1 μM or 0.01 μM. The measurement parameters were as follows: excitation, 620 nm; emission, 670 nm; temperature, 37 °C; acquisition time, 1800 s. Triton X-100 (1%) was used as a control for 100% permeability, and no peptide (0 μM) was the control for 0% permeability. Melittin served as a pore-forming AMP control, and cecropin P1, a carpet-like model AMP control. Each experiment was performed at least twice.

**Outer–inner membrane permeation.** The lactose permease-deficient strain *E. coli* ML35 was cultured in LB broth to log phase and washed in PBS. *E. coli*, peptides, and probes (nitrocefin or ONPG) were diluted in PBS. The peptide (25 μl), probe (25 μl), and *E. coli* (50 μl) were then mixed in 96-well polyethylene plates to the following final concentrations: *E. coli*, OD 0.2; peptides, 40 μM, 20 μM, 10 μM and 5 μM; nitrocefin, 30 μM; ONPG, 2.5 mM. Immediately after mixing, each 96-well plate was placed in a SpectraMax plate spectrophotometer (Molecular Devices, California, USA), and the absorbance was measured. Peptide solution (100 μl) was automatically transferred and mixed with the bacterial solution. The measurement parameters were as follows: absorbance for outer membrane permeation probe (nitrocefin), 490 nm; absorbance for inner membrane permeation probe (ONPG), 420 nm; temperature, 37 °C; acquisition time, 60 min. Melittin was used as a control for permeation of both the inner and outer membrane. Each experiment was performed at least twice.

**Field emission scanning electron microscopy.** FE-SEM was used to observe peptide-induced changes in bacterial morphology. MDR *A. baumannii* (ATCC BAA-1605) and MDR *S. aureus* (KCCM 40510) were cultured in CAMHB to log phase and washed in PBS. Bacteria and peptides were diluted in PBS, after which aliquots of bacteria (50 μl) and peptides (50 μl) were mixed to yield the following final concentrations: MDR *A. baumannii* or MDR *S. aureus*, $2 \times 10^8$ CFU/ml; WCopW29, 2.5 μM. Mixtures were deposited onto cover glass slips and incubated for 0 min, 15 min, 30 min, 1 h or 4 h. After incubation, the bacteria were fixed with paraformaldehyde (4%) and osmium tetroxide (1%) (Sigma, Darmstadt, Germany), washed with water, and freeze-dried overnight. Freeze-dried samples were Pt-coated and placed in an S-4700, EMAX System field emission scanning electron microscope (Hitachi, Tokyo, Japan). Bacterial morphologies were imaged and recorded on video at magnifications of 10,000× and 25,000×.

**Antimicrobial kinetics.** MDR *A. baumannii* (ATCC BAA-1605) and MDR *S. aureus* (KCCM 40510) in midlog phase and peptides were diluted in PBS, after which aliquots of bacteria (50 μl) and peptide solution (50 μl) were mixed in 96-well plates and incubated for 18 h at 37 °C. The final bacteria concentration was $1 \times 10^5$ CFU/ml, and the final peptide concentrations were 2.5 μM, 1.25 μM, 0.63 μM and 0 μM. The mixtures were incubated at 37 °C for 0 min, 15 min, 30 min, 1 h, 2 h, 4 h, 8 h, 16 h or 18 h then collected and spread on MH agar plates. The CFUs were counted after incubation overnight at 37 °C.

Antimicrobial kinetics for each peptide and each bacterial strain were assessed using FE-SEM. The final concentration of bacteria used was $1 \times 10^8$ CFU/ml, and the final peptide concentrations were 20 μM, 10 μM, 5 μM, 2.5 μM and 1.25 μM. Each experiment was performed in triplicate.

**Resistance acquisition.** *A. baumannii* (KCCM 40203) and *S. aureus* (KCTC 1621) were selected as drug-sensitive bacteria. Mutation-susceptible tetracycline (an antibiotic that targets ribosomes) (Sigma, Darmstadt, Germany) and ofloxacin (an antibiotic that targets DNA gyrase) (Sigma, Darmstadt, Germany), mutation-resistant colistin (an antibiotic that targets lipopolysaccharide) (Sigma, Darmstadt, Germany) and vancomycin (an antibiotic that targets peptidoglycan pentapeptide) (Sigma, Darmstadt, Germany) were used as control antibiotics. The MIC for each antibiotic against bacteria (at OD 0.01) was measured. Bacteria were incubated with an antibiotic (×4, ×2, ×1 or ×0.5 MIC) in CAMHB for 18 h at 37 °C. The MIC for each passage was confirmed by additional broth microdilutions under CLSI guidelines. Resistance acquisition was induced for 30 days (30 passages), and each MIC value was confirmed by an independent MIC test against bacteria at each passage.

**In vivo activity tests.** Female BALB/c mice (8-week old; $n = 10$) were purchased from orientbio (Gyeonggido, Republic of Korea) and fed under standardized environmental conditions for 1 week to be used to estimate time-dependent mouse survival rates. After inducing neutropenia by intraperitoneal injection of cyclophosphamide (150 mg/kg; Sigma), MDR *S. aureus* (KCCM 40510), MDR *A. baumannii* (colistin-resistant strain) or MDR *K. pneumonia* (colistin-resistant strain, NCCP 16125) was diluted in PBS and subcutaneously injected into mice (*S. aureus*, $1.5 \times 10^8$ CFU/ml; *A. baumannii*, $1 \times 10^8$ CFU/ml; *K. pneumonia*, $1 \times 10^9$ CFU/ml). After bacterial infection, 100 μl of WCopW43 peptide dissolved in distilled water (10 mg/ml) were injected (50 mg/kg) into each infected site, and survival was monitored every 12 h for 7 days. For *A. baumannii* infection, mice were intraperitoneally injected with mucin (3 mg/ml, 1% in PBS, 300 μl) to suppress macrophages, and *A. baumannii* cells were incubated with mucin (1% in PBS) for 3 h before subcutaneous injection of $2 \times 10^8$ CFU/ml (0.2% mucin in PBS)[71–74]. Survival rates were calculated using the formula: 100% × (number of surviving mice/ten mice). One day after WCopW43 (50 mg/kg) injection, four mice were randomly selected and sacrificed by $CO_2$ inhalation. Solid tissues (60–120 mg) collected from the injection sites were diluted in PBS (1 mg/20 μl) and homogenized using a VCX 500 ultrasonic processor (Sonics, Newtown, USA). Homogenized samples were spread onto MH agar plates, and CFU values were determined after incubation overnight at 37 °C. All animal experiments were carried out with all ethical regulations of Gwangju institute of science and technology animal for animal testing.

**In vivo toxicity tests**. To assess nephrotoxicity, mice (female BALB/c, 8 weeks old; $n = 3$) were subcutaneously administered WCopW43 (100 mg/kg) or PBS (controls) three times (once every 24 h). Blood samples were then collected from the retroorbital veins into heparin-coated glass capillary tubes (Kimble Chase, NJ, USA), which were then centrifuged at 4000 rpm. The serum was collected and analyzed using a blood urea nitrogen (BUN) assay kit (Arbor Assay, Ann Arbor, USA) and a creatinine assay kit (Arbor Assay). Levels of BUN and blood creatinine clearance were measured using a SpectraMax plate spectrophotometer (Molecular Devices) at 450 nm and 490 nm, respectively. All animal experiments were carried out with all ethical regulations of Gwangju institute of science and technology animal for animal testing.

**Supplementary method**. Experimental conditions and methods for maximum-soluble-concentration measurements (Supplementary Table 4), circular dichroism spectrum (Supplementary Fig. 3), artificial membrane calcein permeation measurements (Supplementary Fig. 8), and dynamic light scattering measurements (Supplementary Fig. 9) are described in Supplementary method.

**Statistics and reproducibility**. Unpaired $t$ tests were used to make statistical comparisons with negative controls (PBS-treated). Significance levels were marked as follows: *$P < 0.05$, **$P < 0.01$, ns: not significant. Most tests were performed three times to confirm reproducibility, though tryptophan fluorescence measurements, isothermal titration calorimetry measurements, dynamic light scattering measurements, and resistance acquisition induction were performed twice. In vivo survival rates were assessed once.

**Reporting summary**. Further information on research design is available in the Nature Research Reporting Summary linked to this article.

## Data availability

Analysis of certification of antimicrobial peptides and other raw data are available at Figshare (https://doi.org/10.6084/m9.figshare.20115488) under the name "Matching amino acids membrane preference profile to improve the activity of antimicrobial peptides.zip". All other data are available from the corresponding author on reasonable request.

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

## Acknowledgements

This research was supported by a "GIST Research Institute (GRI) IIBR" grant funded by the GIST in 2022.

## Author contributions

Conception and design: J.K. and S.K. Development of methodology and material support: J.K. and S.K. Acquisition of the data: S.K., J.L., S.L., H.K., J.-Y.S., B.P., and K.K. Obtaining funding and study supervision: Professor J.K. All authors provided final approval of the submitted and published versions of this manuscript.

## Competing interests

The authors declare no competing interests.
