## [Peer Review File · Communications Biology]

Reviewers' comments:

Reviewer #1 (Remarks to the Author):

The authors have submitted a coherent manuscript with sufficient trials meaning a huge effort to show that the rational design of peptides based on the structure of phospholipids by matching amino acids membrane, exhibits a high correlation between interaction with membrane models and antimicrobial activity. However, the results are not entirely comparable between the different trials. Perhaps an adjustment in the design could enhance the manuscript. Thus, some comments should be considered before accepting the manuscript for publication.

Line 21 (page 4): Figures should be cited in the manuscript in a sequential order.

Line 1 (Page 5): Change the verb identify to a more appropriate one.

Line 20 (page 8): It could be better expressed as a peptide-lipid molar ratio.

Line 3 (page 9): It is not understandable how the fluorescence emitted by tryptophan is related to the stability of the MIC. explain better please.

Line 17 (page 10): Explain

Line 6 (page 11): How is the localization of these hydrophobic residues within a bilayer ensured? theoretical studies should be developed.

Line 20 (page 11): There are many interesting results that should be discussed. In itself, the discussion does not explain why the rational design of the sequences, for example, how the addition or substitution of residues on the sequences influences the different variables evaluated such as antimicrobial activity, interaction with membranes, permeability... In addition, the results obtained with murine models should also be discussed. The discussion should be improved.

Line 12 (page 15): Please mention if the peptides were TFA salt or not.

Line 22 (page 15): please mention the proportions used

Line 3 (page 16): Please clarify the preparation method and temperature.

Line 16 (page 16): Express the concentration with exponential

Line 18 (Page 16): Why were ug/ml concentrations not used, considering the CLSI guidelines?

Line 19 (page 16): Was the mic determined visually?

Line 4 (page 17): SEM assays Does not present bacterial OD. What was the criteria to consider different bacterial concentrations?

Line 9 (page 17): in which were the peptides diluted?

Line 16 (page 17): If melittin was purchased, why was it not used as a positive control?. Is Peptides 0ug/ml PBS?

Line 19 (page 17): Why was the mammal model not included?

Line 21 (page 17): The concentrations are confused. It could be expressed as a molar ratio.

Line 12 (page 18): What is the criteria for depositing the peptide in the cell and the lipid in the syringe?

Line 20 (page 18): I suggest that the management of bacteria should be in one section. and membrane models in another section

Line 14 (page 21): Bacterial names should be in italics. Review the entire manuscript

Line 10 (page 22): How was the peptide sterilized? In what solvent was it diluted? What was the concentration of the peptide solution used?
conclusions were not included

Reviewer #2 (Remarks to the Author):

Summary

In this work, the authors rationally designed antimicrobial peptides (AMPs) that are capable of interacting favourably with the bacterial membrane surface, interface, and core region. This was

achieved by considering the corresponding placement of amino acids with different properties along the primary peptide sequence. In their model, linear β -stranded AMPs approximating 10 amino acid residues (wherein charged residues such as Lys/Arg flank both termini, complemented by a middle hydrophobic stretch surrounded by aromatic residues) are proposed to align perpendicularly with bacterial membranes to encourage penetration. WCopW5 (NH₂-wllwiglrkkr-CONH₂) is an AMP derived from insect coprisin with good antimicrobial activity that loosely abides to this model. Here, the authors curated a library of WCopW5 derivatives and successfully demonstrated that linear analogues with primary sequences more closely "matching" the desired membrane alignment pattern are better able to promote bacterial membrane binding, insertion, and permeation in vitro. The lead compound identified from this study (WCopW43, NH₂-rrrwiwlvwk-CONH₂) was tested to be efficacious against a panel of multidrug resistant gram-negative bacteria while demonstrating mammalian safety and sufficient bacterial clearance when administered in mice models.

The authors propose an intriguing and well-conceptualized approach to the rational design of AMPs that takes into consideration membrane properties and amino acid engagement propensity. The impact of this study extends beyond the presented findings to more generally underscore the advantages to be gained from exploring the use of linear AMPs.

Comments

- The authors explore an interesting perspective on the advantages of matching peptide sequence to preferred membrane position to enhance antimicrobial activity. Is it possible for charges attributable to the peptide N- or C-terminus to also contribute a role in this model (i.e., could an exposed amine group on the N-terminus be perceived similarly to Lys/Arg residues)?
- It would be valuable for the authors to elaborate on how a membrane-matched peptide is anticipated to navigate perpendicularly within the membrane when both ends of the primary sequence exhibit a charged residue (ex. in a sequence such as WCopW43, how does one peptide terminus overcome initial electrostatic interactions with the membrane surface to favourably traverse across the bilayer?)
- As this work primarily focuses on the preferred matching of amino acids to the inner bacterial membrane, it becomes curious whether the primary arrangement of residues also inadvertently influences the efficiency of an AMP to cross the outer membrane. Do the authors have any insight on this? Has there been any experiments conducted to compare the OM permeability of membrane-matched and unmatched constructs?
- On Page 9, paragraph 2, it would be worthwhile to briefly comment on the kinetics of membrane permeation for LWCopW29, which seems to rapidly achieve 100% of its fluorescence intensity even when compared to WCopW29. This is consistent with the concept of strong membrane interactions seen for L-amino acids described later on.
- For clarification, is it thought that membrane-matched peptides induce pores in the absence of AMP-AMP interactions as peptide insertion is not shown to decline in the presence of high lipid concentrations?
- Table 1 and 2, likely typo in table heading: change "maximum inhibitory concentrations" to "minimum inhibitory concentrations".
- Just as a suggestion: Tables 1, 2, and 3 seem very large, and perhaps tend to bury the most active peptides in an excess of inactive or less active analogs. Can the authors shorten each of these Tables to highlight the key structures?

Reviewer #3 (Remarks to the Author):

In this ambitious and detailed study the authors derive a new class of antimicrobial peptides based

around common structural features originally observed in derivatives of protegrin-1. Based on careful measurements of the impact of substitutions affecting compatibility with different regions of cell membranes, the group identify a core peptide which is then derivatised further to provide further peptides including one with very high activity both in vitro and in vivo. Although some of the assumptions in the earlier sections appear a little speculative in places, the validity of the approach is borne out by successfully engineering a peptide which has apparent clinical utility.

The paper is unusual in both generating some detailed mechanistic insights, and then utilising these to derive some novel lead compounds with strong activity. The approach involves optimising novel peptides AMPs active in beta-sheet confirmation (which are shorter and potentially more suitable for pharmaceutical development than comparable peptides active in alpha-helical conformation).

The study does have scope to influence thinking in the field both from the perspective of a novel strategy to optimise generic antimicrobial peptide activity, and also the useful compounds that have been identified as a result.

The statistical analysis carried out in the paper was adequate, and it was considerable detail throughout enabling replication if desired.

There were a few minor issues the authors might address:-

- On page 7, comparisons of the activities of peptides AcWL-1, AcWL-2 and AcWL-3, were framed around the positioning of tryptophan residues. However a simpler explanation might be that the most active peptide had two tryptophans, whereas the two less active ones only contained a single tryptophan (given this residue is frequently associated with antimicrobial activity).
- Also on page 7 it was not clear where data for Ptg C-ter, AcWL-3, WCopW5, Hybrid-3 and Hybrid-4 were located. It would be helpful if a reference to the correct Figure or Table was provided here.
- There were a few places where there were minor typos (for example, the opening line of the introduction: 'threat' should be 'threaten') – the manuscript perhaps requires a final proof-read.

In accordance with your and the reviewer's comment, we have carefully revised the manuscript.

The point-by-point responses made to the manuscript are listed below. Please remind that the sentences modified in the manuscript are highlighted in yellow-filled letters, and the location of modified sentences in the manuscript is highlighted with red bolded letters.

The authors have submitted a coherent manuscript with sufficient trials meaning a huge effort to show that the rational design of peptides based on the structure of phospholipids by matching amino acids membrane, exhibits a high correlation between interaction with membrane models and antimicrobial activity. However, the results are not entirely comparable between the different trials. Perhaps an adjustment in the design could enhance the manuscript. Thus, some comments should be considered before accepting the manuscript for publication.

[Comment 1] Line 21 (page 4): Figures should be cited in the manuscript in a sequential order.

[Response 1] According to the comment, we now cited figures in a sequential order.

[old script] Fig. 1b

→ **[revised script] Line 22 (page 4)** Fig. 1

[Comment 2] Line 1 (Page 5): Change the verb identify to a more appropriate one.

[Response 2] According to the comment, we changed to a more appropriate one.

[old script] we identified a derivative of CopW

→ **[revised script] Line 2 (Page 5)** we developed a CopW derivative

[Comment 3] Line 20 (page 8): It could be better expressed as a peptide-lipid molar ratio.

[Response 3] According to the comment, we inserted a peptide-lipid molar ratio.

[old script] The fluorescence of all analogs was increased by increasing lipid concentrations from 0 μM to 100 μM . However, a high lipid concentration of 1000 μM quenched the fluorescence of HLWCopW29 2 and HLWCopW29-4,

→ **[revised script] Line 18 (page 8)** The fluorescence of all analogs was increased by increasing the lipid concentration from a peptide:lipid ratio of 1 μM :0 μM to 1 μM :100 μM . However, a peptide:lipid ratio of 1 μM :1000 μM quenched the fluorescence of HLWCopW29 2 and HLWCopW29-4,

[Comment 4] Line 3 (page 9): It is not understandable how the fluorescence emitted by tryptophan is related to the stability of the MIC. explain better please.

[Response 4] We fixed for better understanding.

[old script] These data, obtained using artificial bacterial membranes, is corresponding with that the MIC of WCopW29 is stable in the presence of high bacteria concentrations (a 2-fold increase of the MIC in the presence of a 100-fold increase in bacterial cell numbers), while the MIC of melittin in the presence of high bacteria concentrations is less stable (a 16-fold increase of MIC) (Fig. S10, Table S2).

→ **[revised script] Line 2 (page 9)** This finding, obtained using artificial bacterial membranes, corresponds to the antimicrobial activities of WCopW29. If WCopW29 and LWCopW29 efficiently insert into an excessive concentration of artificial bacterial membrane, they would also be expected to insert into an excessive concentration of true bacteria membrane. When the *S. aureus* concentration was increased 200-fold, LWCopW29 and WCopW29 MIC were increased only 2-fold, while HLWCopW29-2

and melittin MIC were increased 8-16-fold (Fig. S10, Table S2).

[Comment 5] Line 17 (page 10): Explain

[Response 5] Exposure of bacteria to sub-lethal concentrations of antibiotics is an artificial method to induce antibiotic-resistant bacteria. However, antimicrobial peptides activities are stable against the emergence of antibiotic-resistant bacteria because antimicrobial peptides target lipid charge. The probability of resistance is low because the fitness cost for a microbe to reconfigure its entire membrane is high (manuscript ref. 68).

[Comment 6] Line 6 (page 11): How is the localization of these hydrophobic residues within a bilayer ensured? theoretical studies should be developed.

[Response 6]

We did not observe the allocation in a real membrane, so we cannot be certain that the hydrophobic position-preferring residues (WIWVLW) were actually situated within the hydrophobic core and that the hydrophilic position-preferring residues (NH₂, CONH₂, RRR, K) were actually situated at the surface. Instead, we deduced the membrane binding site from two indirect evidences.

→ **[revised script] Line 17 (page 14)** First, the conserved MIC after glycosylation of WCopW47, 48 indicates that neither terminal was a membrane-binding residue (74). The only remaining candidates for the membrane-binding residues are the hydrophobic position-preferring residues.

Second, the peptide-membrane interaction data (Fig. 2, S8, S9) indicates that WCopWs induce membrane poration (70). There are two membrane poration models: the transmembrane model and the interfacial activity model. In the transmembrane model, hydrophobic position preferring residues are located within a hydrophobic core. In the interfacial activity model, all residues of the peptide are evenly located around an interface (68, 70). Thus, if WCopWs do not correspond to the interfacial activity model, it can be concluded that the hydrophobic position preferring residues are indeed located within the hydrophobic core. In the interface activity model, changing the hydrophobic position-preferring residues from L-amino acids to D-amino acids would not induce a significant change in activity (68). By contrast, with our WCopWs, D- to L-amino acid substitution of hydrophobic position-preferring residues (WCopW65) eliminated the peptide's activity. Additionally, single D- to L-amino acid substitution of hydrophilic position preferring residues did not induce a significant change in the activities of WCopW55, 62, 63, 64. This suggests that the location of the hydrophobic and hydrophilic position-preferring residues are not even. The only existing model consistent with those observations without conflict is the transmembrane poration model. (If we had performed the tryptophan quenching experiment with WCopW47, 48 and WCopW65 as well as with the glucosylated hydrophobic position-preferring residue analog or the single D- to L-amino acid substituted hydrophobic position-preferring residue analog, which exhibited a greatly deteriorated MIC, our evidence would have been reinforced, but, unfortunately, we also did not do that experiment.)

[Comment 7] Line 23 (page 11): There are many interesting results that should be discussed. In itself, the discussion does not explain why the rational design of the sequences, for example, how the addition or substitution of residues on the sequences influences the different variables evaluated such as antimicrobial activity, interaction with membranes, permeability... In addition, the results obtained with murine models should also be discussed. The discussion should be improved.

[Response 7] While trying to answer for revision, we realized that some of our answers should have been included in the

discussion. Therefore, we thought that the discussion should be improved by organizing those answers in the discussion. This part includes an answer to the comment for [Comment 6] Line 6 (page 11) and [Comment 17] Line 19 (page 17) of the first reviewer and [Comment 1], [Comment 2], [Comment 4], and [Comment 5] of the second reviewer.

→ [revised script] Line 23 (page 11) In this study, we developed colistin- and daptomycin-comparable AMPs through rational design. Thanks to advances in screening methods, a number of potent AMPs have been acquired through cost-effective and creative random screening (2, 66). On the other hand, AMPs rationally designed based on advanced understanding of AMPs were less effective than AMPs generated through repetitive trial-and-error (28, 62). This is confusing, given that the structures of the rationally designed AMPs were based on an extensive understanding of the complex membrane-peptide interactions. Nevertheless, we believe that rational understanding can provide guidance that cannot be obtained in other ways. Occasionally, we obtained unique AMPs like the WCopWs, the activity of which cannot be explained by the canonical structure-activity relationship of AMPs. However, extensive and detailed comparison of the structure-activity relationships of sequentially similar peptides, membrane-peptide interaction data, and the site-directed modified analogs guided us to optimize the sequences of AMPs by matching the amino acid preferred position profile of the membrane. This profile matching guided us to rationally design WCopW43, which is significantly more potent than previously described nonclinical AMPs. This rational approach also guides further study. For example, we could try to conjugate vancomycin or colistin to surface position-preferring residues to produce a synergistic effect (69) without disrupting peptide-membrane insertion. Profile matching can be more perfectly optimized by modulating backbone length with various peptidomimetics like β -amino acids (7). If an accurate amino acid preferred position profile for lipopolysaccharide or cholesterol was obtained, a more membrane-selective candidate could be designed. A preferred position profile could also be applied to the development of cell-penetrating peptides (CPPs). Although the similarity between AMPs and CPPs is well known, profile-matched peptides are more closely correlated with CPPs when considering that primary amphipathy is the major common property of CPPs. With a simple idea and few technical requirements, preferred position profile-matched peptides have the potential for extensive applicability.

Although it was not a focus of this study, β -strand conformation is also important for the activity of WCopWs (Fig. S1). Strong activity correlated with the appearance of a negative 218 nm circular-dichroism peak (Fig. S4). It is known that an extended β -strand conformation is essential for construction of a pore structure by 9- or 10-mer peptides (26, 37, 39, 58). Loss of activity caused by introducing L- and D-amino acid repeats into WCopW65, thereby disrupting its secondary structure, also demonstrate the importance of the β -strand conformation (68). β -strand may be as important as preferred position profile matching because the interstrand hydrogen bonding of β strands could provide a stabilizing force for the linear structure from the side, while profile matching could provide a stabilizing force at both terminals.

Earlier studies have suggested that β -strand AMPs have advantages over α -helical AMPs. Phospholipids do not have a straight, cylinder geometry; instead, differences in head and tail diameters lead to a conical geometry (4, 38). The “wedge” (4) or “void” space (38) between cones could facilitate the insertion of AMPs into the membrane core by bypassing hydrophobic-hydrophilic repulsion and lateral pressure. Theoretically, therefore, AMPs whose amino acid sequences are designed with β -strand to optimize membrane insertion are more advantageous than α -helical AMPs because the thin β -strand inserts more easily into the void space (Fig. S1) (38).

Longer length with fewer residues is another advantage that β -strand AMPs have over α helical peptides. Membrane spanning length is important for α -helical amphipathic AMP activity. However, α -helical amphipathic AMPs are limited by their high molecular weight (7), which cannot be avoided because longer sequences are needed for the stable formation of α -helical

amphipathic structures that span the membrane. If the molar concentration is taken into consideration, the antimicrobial activities of the most active α -helical amphipathic AMPs (such as melittin) are comparable to that of WCopW5 (Table. S1). However, melittin (2.85 kDa) is two times larger than WCopW5 (1.47 kDa), so its actual activity (3.56 $\mu\text{g/ml}$) is half that of WCopW5 (1.84 $\mu\text{g/ml}$), as antimicrobial activity is not expressed in terms of molar concentration but as weight/volume concentration (10).

However, the key feature of β -strand AMPs over α -helical AMPs is clearly their compatibility with the preferred position profile. It is noteworthy that regardless of how we optimize α -helix peptides to the preferred position profile, its most stable state is a surface binding state. For most α -helical AMPs, the transmembrane state is merely an intermediate that occurs when translocating from an outer leaflet surface-bound state to an inner leaflet surface-bound state (3, 70). This is the reason that most α -helical AMP-induced pores (except a few true poration AMPs, such as melittin (70), or β -helical AMPs, such as gramicidin A (71)) are prone to be transient. By contrast, by matching their sequence to the preferred position profile, β -strand AMPs can be stabilized in a transmembrane state. Although we lack direct observation data, we expect that this stable transmembrane state would be the most advantageous property of LWCopW29 and WCopW29 over HWCopW29-1,2,3,4. Our profile matching AMP design has some critical limitations. Although we designed the AMP sequences to match the preferred position profile, we did not observe the allocation in a real membrane, so we cannot be certain that the hydrophobic position-preferring residues (WIWVLW) were actually situated within the hydrophobic core and that the hydrophilic position-preferring residues (NH₂, CONH₂, RRR, K) were actually situated at the surface. Instead, we deduced their location from two pieces of indirect evidence. First, the conserved MIC after glycosylation of WCopW47, 48 indicates that neither terminal was a membrane-binding residue (74). The only remaining candidates for the membrane-binding residues are the hydrophobic position-preferring residues. If we had performed the tryptophan quenching experiment with WCopW47, 48 as well as with the glucosylated hydrophobic position-preferring residue analog, which exhibited a greatly deteriorated MIC, our evidence would have been reinforced, but, unfortunately, we did not do that experiment.

Second, the peptide-membrane interaction data (Fig. 2, S8, S9) indicates that WCopWs induce membrane poration (70). There are two membrane poration models: the transmembrane model and the interfacial activity model. In the transmembrane model, hydrophobic position preferring residues are located within a hydrophobic core. In the interfacial activity model, all residues of the peptide are evenly located around an interface (68, 70). Thus, if WCopWs do not correspond to the interfacial activity model, it can be concluded that the hydrophobic position preferring residues are indeed located within the hydrophobic core. In the interface activity model, changing the hydrophobic position-preferring residues from L-amino acids to D-amino acids would not induce a significant change in activity (68). By contrast, with our WCopWs, D- to L-amino acid substitution of hydrophobic position-preferring residues (WCopW65) eliminated the peptide's activity. Additionally, single D- to L-amino acid substitution of hydrophilic position preferring residues did not induce a significant change in the activities of WCopW55, 62, 63, 64. This suggests that the location of the hydrophobic and hydrophilic position-preferring residues are not even. The only existing model consistent with those observations without conflict is the transmembrane poration model. (If we had performed the tryptophan quenching experiment with WCopW47, 48 and WCopW65 as well as with the glucosylated hydrophobic position-preferring residue analog or the single D- to L-amino acid substituted hydrophobic position-preferring residue analog, which exhibited a greatly deteriorated MIC, our evidence would have been reinforced, but, unfortunately, we also did not do that experiment.)

We also want to discuss the "missing links" in what we know about the mechanism. We observed and compared the binding, insertion, permeation and activity of the tested AMPs. However, we lack data on the events between insertion and permeation and between permeation and activity. To link insertion and permeation, we will need greater insight into the AMP-lipid

interaction, including pore structure and pore life span. Such information could be gained through the use of molecular dynamics simulations or nuclear magnetic resonance analyses. Without those data, for now, we must make inferences based on our observation as well as earlier reported work. One interesting observation was that permeation by LWCopW29 was similar to that of HLWCopW29 when the peptide:lipid ratio was high. This is most likely because at high peptide:lipid ratios surface-binding AMPs (HLWCopW29-2,4) can induce permeation as effectively as transmembrane pore-inducing AMPs (LWCopW29, WCopW29) (70). We also suspect that micellization of LWCopW29 or WCopW29 may limit permeation at high concentrations because primary amphipathic peptides tend to micellize themselves at high concentrations (72). Another interesting observation was that insertion and permeation by LWCopW29 were consistently better than by WCopW29. This is most likely because normal peptides composed of L-amino acids tend to have stronger affinity for the normal liposomes, while peptides composed of D-amino acids tend to have a stronger affinity for similarly enantiomeric liposomes (52). Presumably, the better activity of WCopW29 relies solely on protease stability (Fig. S12). Finally, there is a lack of proportionality between permeability and activity. At 10 μ M, melittin induces more than 3 times as much permeation as LWCopW29. Despite this significantly lower level of permeation, LWCopW29 completely inhibits bacteria but melittin does not (Table S2). The easiest explanation for this result is dual-targeting of LWCopW29. However, the conserved and broad antibacterial spectrum observed with racemized, reverted, and point mutated LWCopW29 analogs is sufficient to reject the possibility of dual-targeting. Consequently, at present we cannot explain this inconsistency. Indeed, the lack of proportionality between permeability and activity is the most common and critical obstacle to a rational approach to AMP development. For example, daptomycin does not induce leakage of fluorescent probes, though it is a strong antimicrobial agent (60). There are many examples of AMP derivatives with permeation activities that are disproportionate with respect to their antimicrobial activities (22, 62, 68). But there are also many examples where these activities are proportionate (46, 49, 63). One possible explanation for the lack of proportionality between permeation and antimicrobial efficacy is negative feedback inhibition of AMPs by leaked cytosolic contents. Macromolecules and salts rapidly released by AMP-induced membrane poration may suppress AMP activity by disrupting electrostatic interactions by binding AMPs or by inducing stress responses (64, 65). If so, considering that AMPs do not need the same level of permeability as melittin because bacterial metabolism can be critically disrupted by partial membrane permeation (73), we would not need to pursue a peptide with marked permeability. Instead, AMPs that can stably induce moderate permeation under challenging conditions (e.g., presence of protease, low peptide:lipid ratio, low residue number, or high salt concentration) may be pursued as a goal of rational design of potent AMPs.

Murine models discussion:

As seen in Table S6, the *A. baumannii* strain used for the mouse model was colistin-resistant bacteria. We intended to use colistin as a negative control and highlight in vivo activity of WCopW43.

However, colistin was effective against colistin-resistant *A. baumannii* strains in a mouse model (survival rate: 90%). (When the MIC test was performed again by obtaining colony from the mouse body, it was still colistin-resistant.)

WCopW43 exhibits only a slightly better survival rate (100%). It's difficult and dangerous to say whether WCopW43 is superior to colistin or equivalent. We also couldn't find a reference to explain this result. So we decided not to compare colistin and WCopW43 with data acquired in the mice model.

[Comment 8] Line 12 (page 15): Please mention if the peptides were TFA salt or not.

[Response 8] We used both of TFA and acetate salt. The script was added.

→ **[revised script] Line 3 (page 19)** All WCopW analogs and cecropin P1, melittin and protegrin-1 were prepared as TFA salts except the WCopW43 used in the in vivo test, which was an acetate salt.

[Comment 9] Line 22 (page 15): please mention the proportions used

[Response 9] According to the comment, the script was fixed.

[old script] Bacterial-membrane-mimic DMPC/DMPG unilamellar vesicles (LUV) and mammalian-membrane-mimic PC/cholesterol

→ **[revised script] Line 9 (page 19)** Bacterial-membrane-mimic DMPC/DMPG (7/3) unilamellar vesicles (LUV) and mammalian-membrane-mimic DMPC/cholesterol (10/1)

[Comment 10] Line 3 (page 16): Please clarify the preparation method and temperature.

[Response 10] According to the comment, we clarified the preparation method and temperature.

[old script] vesicles with 100-nm diameters were prepared by extrusion prepared by extrusion.

→ **[revised script] Line 11 (page 19)** vesicles with 100-nm diameters were prepared by extrusion prepared by freeze-thaw and extrusion. Lipids are mixed and dissolved by methanol in round-bottom-flask and evaporated and freeze-dried for overnight to be a thin layer covering the glass wall. PBS was added to the lipid layer. PBS-lipid layer was vortexed and freeze-thawed by 50° C water and lipid-nitrogen until no visible thin layer remained.

[Comment 11] Line 16 (page 16): Express the concentration with exponential

[Response 11] According to the comment, we fixed the wrong word.

[old script] 5 x 10⁵ colony forming units (CFU)/ml;

→ **[revised script] Line 7 (page 20)** 5 x 10⁵ colony forming units (CFU)/ml;

[Comment 12] Line 18 (Page 16): Why were ug/ml concentrations not used, considering the CLSI guidelines?

[Response 12]

In our thought, using ug/ml concentration benefits in determining whether passing or failing of product.

But, using molar concentration benefits in evaluating, comparing, and anticipating the effect of residual modification on each molecule which would be intermediate and hint to generating a better product because it is more generous about the MW-increasing modifications. (in comparison, using ug/ml concentration could discourage MW-increasing modifications like lipidation, glycosylation, or Gly>Trp substitution)

Therefore, We thought that using molar concentration is more suitable for our AMPs development.

[Comment 13] Line 19 (page 16): Was the mic determined visually?

[Response 13] MIC determined visually. We inserted a script to explain.

[old script] The MIC was defined as the lowest concentration of the peptide that inhibited visible growth of the tested bacteria.

→ **[revised script] Line 7 (page 20)** The MIC was defined to be the lowest concentration of peptide without visible turbidity by comparing with control antibiotics with the unaided eye. Colistin and melittin served as controls for gram-negative bacteria, while daptomycin, vancomycin and melittin were used as controls for gram-positive bacteria.

[Comment 14] Line 4 (page 17): SEM assays Does not present bacterial OD. What was the criteria to consider different bacterial concentrations?

[Response 14] We understand this comment as requiring the reason for using CFU/ml to count bacteria for some experiments (SEM, in vivo) while using OD600 for some experiments (Bacteria membrane ion permeation measurements, Outer-inner membrane permeation)

As we understand, most experiments tend to use absorbance or fluorescence change and also use OD600 to count bacteria. Other experiments tend to use CFU/ml. But SEM doesn't use absorbance or fluorescence change. So we used CFU/ml rather than OD600 to count bacteria.

[Comment 15] Line 9 (page 17): in which were the peptides diluted?

[Response 15] PBS was used. The script was added.

[old script] Human red blood cells (Zenbio, North Carolina, USA) were diluted in PBS (blood concentration, 8%).

→ **[revised script] Line 3 (page 21)** Human red blood cells (Zenbio, North Carolina, USA) and peptides were separately diluted in PBS (blood concentration, 8%).

[Comment 16] Line 16 (page 17): If melittin was purchased, why was it not used as a positive control?. Is Peptides 0ug/ml PBS?

[Response 16] Melittin is used as a positive control in every hemolysis assay. The script was fixed.

[old script] Hemolysis was controlled by including samples with 1% Triton X-100 (for 100% hemolysis) and samples with 0 μ M peptide (for 0% hemolysis)

→ **[revised script] Line 11 (page 21)** Hemolysis was controlled by including samples with 1% Triton X-100 (for 100% hemolysis) and samples with 0 μ M peptide (for 0% hemolysis) and melittin (for positive control).

[Comment 17] Line 19 (page 17): Why was the mammal model not included?

[Response 17] Our theory is based on the AcWL5 transmembrane structure observed in the phosphatidylcholine membrane, the protegrin transmembrane structure observed in the phosphatidylethanolamine/phosphatidylglycerol membrane, and the five-slab model describing the membrane composed with phospholipids. Therefore, the membrane containing cholesterol is just not included in our theory.

To the best of our knowledge, the AcWL5 transmembrane structure is observed in only in the phosphatidylcholine membrane. Protegrins are repelled to a surface binding state in the cholesterol-containing membrane. Five-slab is not matched with cholesterol.

Although the mammal model can give certain valuable insights such as the prediction about hemolysis activity, we thought the primary goal for AMPs development is antimicrobial activity. We thought we can efficiently find less hemolytic AMPs after finding and modifying the template of AMPs with potent antimicrobial activity. So we didn't insight into the mammal model.

[Comment 18] Line 21 (page 17): The concentrations are confused. It could be expressed as a molar ratio.

[Response 18] According to the comment, the confused word was fixed.

[old script] samples with 0 $\mu\text{g/ml}$ peptide

→ [revised script] Line 11 (page 21) samples with 0 μM peptide

[Comment 19] Line 12 (page 18): What is the criteria for depositing the peptide in the cell and the lipid in the syringe?

[Response 19] We caught this comment meaning : Compare the Peptide(syringe)-into-lipid(cell) (PS-LC) and the Lipid(syringe)-into-peptide(cell) (LS-PC)

We used Lipid(syringe)-into-peptide(cell) (LS-PC) by following the methods of

[Thermodynamics of the interactions of tryptophan-rich cathelicidin antimicrobial peptides with model and natural membranes]:

'The peptide solution (10–50 μM , depending on the peptide affinity to the lipid vesicles) was placed in the calorimeter cell. The LUV suspension (10–14 mM; 500 μL) was placed in the titration syringe and injected in aliquots of 15 μL '

[Membrane Binding and Pore Formation of the Antibacterial Peptide PGLa: Thermodynamic and Mechanistic Aspects]: 'In this type of experiment, small aliquots of a concentrated vesicle solution (5–8 μL of 25–50 mM lipid suspensions) are injected into the calorimeter cell containing the peptide solution (between 3 and 40 μM PGLa).'

The reason for choosing LS-PC rather than PS-LC is shown in

[Beyond electrostatics: Antimicrobial peptide selectivity and the influence of cholesterol-mediated fluidity and lipid chain length on protegrin-1 activity]: 'Corollary peptide-into-lipid (P-L) titrations were performed by the repeated injection of 2–3 μL aliquots of a 150–200 μM PG-1 solution into a suspension of LUVs having a total PC concentration of 4–5 mM.'

[Magainin 2 Amide Interaction with Lipid Membranes: Calorimetric Detection of Peptide Binding and Pore Formation]: 'In the first type of experiment, the calorimeter cell contained sonified vesicles composed of POPC and POPG, and small aliquots of M2a were injected'

Those papers compared PS-LC and LS-PC methods. However, the PS-LC method cannot produce usable data. ΔH of each injection is not changed. With those data, K_d is difficult to calculate.

Presumably, 'criteria for depositing the peptide in the cell and the lipid in the syringe' is the capacity of a liposome.

Liposome-peptide interaction is different with peptide-peptide interaction because of this capacity. After certain peptides are inserted into a liposome, those peptides are excluded from another liposome. Even if new liposomes are injected (LS-PC), inserted peptides cannot react with another liposome.

However, after certain liposomes uptake peptides, that liposome can uptake another peptide. Even if new peptides are injected (PS-LC), that liposome can uptake new peptides robustly.

Therefore, LS-PC would be more suitable for acquiring K_d and other peptide-liposome interactions.

[Comment 20] Line 20 (page 18): I suggest that the management of bacteria should be in one section. and membrane models in another section

[Response 20] MIC50, 90 part is separated into two parts and inserted in 'Bacteria preparation' and 'MIC measurement' respectively. (please see Line 10 (page 18) and Line 20 (page 20) to check change)

[Comment 21] Line 14 (page 21): Bacterial names should be in italics. Review the entire manuscript

[Response 21] According to the comment, typeface was fixed.

A. baumannii → *A. baumannii*: **Line 16, 26 (page 38)**

S. aureus → *S. aureus*: **Line 1 (page 24), Line 12, 14 (page 25), Line 6 (page 29), Line 23 (page 34), Line 23, 25 (page 37), Line 26 (page 38)**

K. pneumonia → *K. pneumonia*: **Line 26 (page 38)**

[Comment 22] Line 10 (page 22): How was the peptide sterilized? In what solvent was it diluted? What was the concentration of the peptide solution used?

[Response 22] Peptides are not sterilized. Distilled water was used for in vivo test (because we observed turbid in high concentration (10 mg/ml) of WCopW43 peptide in PBS). We added sentence.

Line 17 (page 25) → After bacterial infection, the 100 µl of WCopW43 peptide dissolved in distilled water (10 mg/ml) was injected (50 mg/kg) into each infected site, and survival was monitored every 12 h for 7 days.

conclusions were not included

Reviewer #2 (Remarks to the Author):

Summary

In this work, the authors rationally designed antimicrobial peptides (AMPs) that are capable of interacting favourably with the bacterial membrane surface, interface, and core region. This was achieved by considering the corresponding placement of amino acids with different properties along the primary peptide sequence. In their model, linear β -stranded AMPs approximating 10 amino acid residues (wherein charged residues such as Lys/Arg flank both termini, complemented by a middle hydrophobic stretch surrounded by aromatic residues) are proposed to align perpendicularly with bacterial membranes to encourage penetration. WCopW5 (NH₂-wllwigrkkr-CONH₂) is an AMP derived from insect coprisin with good antimicrobial activity that loosely abides to this model. Here, the authors curated a library of WCopW5 derivatives and successfully demonstrated that linear analogues with primary sequences more closely “matching” the desired membrane alignment pattern are better able to promote bacterial membrane binding, insertion, and permeation in vitro. The lead compound identified from this study (WCopW43, NH₂-rrrwiwlwk-CONH₂) was tested to be efficacious against a panel of multidrug resistant gram-negative bacteria while demonstrating mammalian safety and sufficient bacterial clearance when administered in mice models.

The authors propose an intriguing and well-conceptualized approach to the rational design of AMPs that takes into consideration membrane properties and amino acid engagement propensity. The impact of this study extends beyond

the presented findings to more generally underscore the advantages to be gained from exploring the use of linear AMPs.

Comments

[Comment 1] • The authors explore an interesting perspective on the advantages of matching peptide sequence to preferred membrane position to enhance antimicrobial activity. Is it possible for charges attributable to the peptide N- or C-terminus to also contribute a role in this model (i.e., could an exposed amine group on the N-terminus be perceived similarly to Lys/Arg residues)?

[Comment 2] • It would be valuable for the authors to elaborate on how a membrane-matched peptide is anticipated to navigate perpendicularly within the membrane when both ends of the primary sequence exhibit a charged residue (ex. in a sequence such as WCopW43, how does one peptide terminus overcome initial electrostatic interactions with the membrane surface to favourably traverse across the bilayer?)

[Response 1, 2] We understand that the first and second comments as a question for 'why preferred position profile matched AMPs are more favored than helical amphipathic AMPs even though preferred position profile matched AMPs also require traverse of charge molecules (NH₂, -Lys-CONH₂). Therefore, we think it would be appropriate to answer at once.

As indicated, although our indication about the irony of helical AMPs is the energy penalty of charged residues to traverse across the bilayer, preferred matched peptides also require the traverse of charged residues.

However, in our thought, preferred matched peptides still suffer less energy penalty than helical AMPs because helical AMPs require insertion of 'all' charged residues distributed on the entire sequence, while preferred matched peptides need to traverse 'part' of charged residues at one end.

For example, in the case of HWCopW29, it should transport all +4 charged residues from bilayer outer leaflet surface to the inner leaflet surface (according to AMP-pore ensemble model or partial transient release model). Meanwhile, WCopW29 need to transport only part of charged residues (+1 charged residue(NH₂-), If a WCopW29 transmembrane state is a parallel, similar to a protegrin transmembrane structure. Even if a WCopW29 transmembrane state is antiparallel, similar to AcWL5, it is still energetically favored than HWCopW29 because transporting +3 charged residue (-Arg-Arg-Arg-CONH₂) is more favored than transporting +4 charged molecule (HWCopW29))

(Additionally, we think that a beta-strand structure can reduce the energy penalty. please see **line 3, page 13** in the improved discussion of manuscript).

[Comment 3] • As this work primarily focuses on the preferred matching of amino acids to the inner bacterial membrane, it becomes curious whether the primary arrangement of residues also inadvertently influences the efficiency of an AMP to cross the outer membrane. Do the authors have any insight on this? Has there been any experiments conducted to compare the OM permeability of membrane-matched and unmatched constructs?

[Response 3] Unfortunately, We have not conducted experiments to insight the difference between membrane-matched peptides interacting with the outer membrane and unmatched peptides interacting with the outer membrane.

Although we were thinking that linear peptides were likely to more selectively interact with the outer membrane (Especially, we were speculating that there would be a link between outer-membrane-porin (OMP) and beta-strand structure and outer membrane selectivity), making an artificial outer-membrane to insight and prove such thinking was impossible with our device (Also, to the best of our knowledge, there are no OMP's which has a similar sequence to WCopWs.).

Therefore, experiments such as Figure 2 are hardly possible. Without insertion and binding data, we thought it was difficult to compare the difference between membrane-matched peptides and unmatched peptides only with permeation data. So we didn't compare the OM permeability of membrane-matched and unmatched constructs.

(please see 'Revision tables and figures. pdf') So, we have done a simple 'outer-inner membrane disruption-permeation' experiment once, and we are not sure what the data and presume obtained from this experiment. We think we need to increase the quantity and quality of the experimental data through additional experiments, but we think it will take more time and effort.

[Comment 4] • On Page 9, paragraph 2, it would be worthwhile to briefly comment on the kinetics of membrane permeation for LWCopW29, which seems to rapidly achieve 100% of its fluorescence intensity even when compared to WCopW29. This is consistent with the concept of strong membrane interactions seen for L-amino acids described later on.

[Response 4] Reference 52, 'Is the Mirror Image a True Reflection? Intrinsic Membrane Chirality Modulates Peptide Binding' indicates that 'chirality of lipid bilayers can modulate peptide–lipid interactions'

We found other references presenting better interaction between L-peptide with natural liposome than D-peptide.

'(Peptide with) L-amino acids show a stronger affinity for the liposomes compared to the ones with D-amino acids' [Chiral Recognition of Lipid Bilayer Membranes by Supramolecular Assemblies of Peptide Amphiphiles. Sato K, et al. ACS Biomater Sci Eng. 2019.]

'(On natural lipid membrane), The L-peptide enantiomer absorbed easily, but the D-peptide enantiomer had to overcome an extra free-energy barrier' [Probing the Role of Chirality in Phospholipid Membranes. Martin HS, et al. Chembiochem. 2021]

Therefore, the matching of lipid chirality and peptide chirality looks like playing a major role in the stronger membrane interactions of LWCopW29 than WCopW29 (please see **line 14, page 16** in the improved discussion of manuscript).

[Comment 5] • For clarification, is it thought that membrane-matched peptides induce pores in the absence of AMP-AMP interactions as peptide insertion is not shown to decline the in presence of high lipid concentrations?

[Response 5]

Unfortunately, we cannot state that 'matched peptides induce pores in the absence of AMP-AMP interactions'. We should speak more carefully that 'matched peptides induce pores with relatively fewer AMP-AMP interactions than unmatched AMPs'. There is one analog to consider; WCopW65. D- to L- substituted analog, WCopW65, completely lost its activity. D- to L- substitution disrupts AMP-AMP interaction. Therefore, AMP-AMP interaction has a role in pore-inducing. (please see **line 3, page 13** in the improved discussion of manuscript).

What is the role of 'AMP-AMP interaction' in 'pore-inducing of profile matched AMPs'? Two hypotheses can be established.

First: ' β -strand structure (AMP-AMP interaction)' is required for transmembrane insertion. An Interstrand hydrogen bond between at least two peptides is required for stabilizing the linear transmembrane structure.

Second: ' β -strand structure' is not required for transmembrane insertion. It is required after insertion to assemble and construct a pore structure.

Both hypotheses are not conflicting with our data and model. We can't choose between the two hypotheses.

If we had measured the tryptophan quenching data of WCopW65, it could be an important hint. If WCopW65 is not as well inserted as WCopW29, the first hypothesis is correct. If inserted well, the second hypothesis is correct. But unfortunately, we did not measure.

[Comment 6] • Table 1 and 2, likely typo in table heading: change “maximum inhibitory concentrations” to “minimum inhibitory concentrations”.

[Response 6] That typo is a very critical mistake. Thanks for the indication.

[Comment 7] • Just as a suggestion: Tables 1, 2, and 3 seem very large, and perhaps tend to bury the most active peptides in an excess of inactive or less active analogs. Can the authors shorten each of these Tables to highlight the key structures?

[Response 7] Definitely, there are some less necessary analogs, especially in table 1. We have re-selected analogs in Tables 1 and 3.

Reviewer #3 (Remarks to the Author):

In this ambitious and detailed study, the authors derive a new class of antimicrobial peptides based around common structural features originally observed in derivatives of protegrin-1. Based on careful measurements of the impact of substitutions affecting compatibility with different regions of cell membranes, the group identify a core peptide which is then derivatised further to provide further peptides including one with very high activity both in vitro and in vivo. Although some of the assumptions in the earlier sections appear a little speculative in places, the validity of the approach is borne out by successfully engineering a peptide which has apparent clinical utility.

The paper is unusual in both generating some detailed mechanistic insights, and then utilising these to derive some novel lead compounds with strong activity. The approach involves optimising novel peptides AMPs active in beta-sheet confirmation (which are shorter and potentially more suitable for pharmaceutical development than comparable peptides active in alpha-helical conformation).

The study does have scope to influence thinking in the field both from the perspective of a novel strategy to optimise generic antimicrobial peptide activity, and also the useful compounds that have been identified as a result.

The statistical analysis carried out in the paper was adequate, and it was considerable detail throughout enabling replication if desired.

There were a few minor issues the authors might address:-

[Comment 1] • On page 7, comparisons of the activities of peptides AcWL-1, AcWL-2 and AcWL-3, were framed around the positioning of tryptophan residues. However a simpler explanation might be that the most active peptide had two tryptophans, whereas the two less active ones only contained a single tryptophan (given this residue is frequently associated with antimicrobial activity).

[Response 1] In our thought, MIC difference (MIC of AcWL-1 against *A.baumannii*: 2.5uM. MIC of AcWL-2 against *A.baumannii*: 10uM) between peptides (AcWL-1: WLLLLRRR, AcWL-2: LLLLWRRR) with same trp composition could support the importance of tryptophan positioning.

(Although we cannot deny the tendency which aromatic residues are not too position-sensitive as charged residues. As marked as 'ironically' in the abstract and introduction, the most influential factor for antimicrobial activity is the position of the charged amino acids.)

[Comment 2] • Also on page 7 it was not clear where data for Ptg C-ter, AcWL-3, WCopW5, Hybrid-3 and Hybrid-4 were located. It would be helpful if a reference to the correct Figure or Table was provided here.

[Response 2] Certainly, it is confusing. So we provided additional references AND a sequence of samples for better understanding as follows. Would it be allowable as it is?

[old script]

For the peptide to extend perpendicularly and linearly across the membrane without a stabilizer, the conformation must be stabilized by balancing the location of the aromatic-ring amino acid and the cationic amino acid at both ends of hydrophobic carbon-chain amino acids (AcWL-1 and AcWL-3), not at one end (AcWL-2).

The absence of this balanced location explains the weak activity of protegrin truncates (Ptg C-ter 1, 2, 3) (35, 36). However, AcWL-1 and AcWL-3 have hemolysis activity and are insoluble, presumably due to their high content of hydrophobic leucine (Table 2, Table. S4) (2).

→ **[revised script] (Line 5, Page 7)** → For a peptide to extend perpendicularly and linearly across a membrane without a stabilizer, the conformation must be stabilized by balancing the location of an aromatic-ring amino acid and a cationic amino acid on both sides of the hydrophobic carbon chain amino acids (AcWL-1 (WLLLLRRR) and AcWL-3 (WLLLWRRR)), not just on one side (AcWL-2 (LLLLWRRR)). The absence of this balanced location explains the weak activity of truncated protegrin analogs (Ptg C-ter 1, 2, 3) (35, 36). However, AcWL-1 and AcWL-3 exhibited hemolytic activity and were insoluble, presumably due to their high content of hydrophobic leucine (Table 2, Table. S4) (2).

[Comment 3] • There were a few places where there were minor typos (for example, the opening line of the introduction: 'threat' should be 'threaten') – the manuscript perhaps requires a final proof-read.

[Response 3] Thanks for the indication. We proofread the manuscript ourselves and expert.

REVIEWERS' COMMENTS:

Reviewer #1 (Remarks to the Author):

Suggestions were satisfactorily addressed by the authors and the manuscript improved substantially. The manuscript can be published.

Reviewer #2 (Remarks to the Author):

In their revised manuscript, the authors have responded systematically to my reviewer's comments/concerns, and with conceptually thoughtful commentary. As well, while not claiming to have gone through their responses to the other reviewers point-by-point, it appears that the authors have largely succeeded in addressing their comments. Therefore, assuming the other reviewers concur, I am able to recommend publication of this manuscript in its present form. The final draft of the paper may need attention to editing to ensure journalistic English.

Reviewer #3 (Remarks to the Author):

I believe that my previous comments have been addressed.